# Regulating the electronic structure through charge redistribution in dense single-atom catalysts for enhanced alkene epoxidation

Hongqiang Jin [1,2,5], Kaixin Zhou [1,2,5], Ruoxi Zhang [1,2], Hongjie Cui[1], Yu Yu [3] ✉, Peixin Cui [4] ✉, Weiguo Song [1,2] & Changyan Cao [1,2] ✉

Inter-site interaction in densely populated single-atom catalysts has been demonstrated to have a crucial role in regulating the electronic structure of metal atoms, and consequently their catalytic performances. We herein report a general and facile strategy for the synthesis of several densely populated single-atom catalysts. Taking cobalt as an example, we further produce a series of Co single-atom catalysts with varying loadings to investigate the influence of density on regulating the electronic structure and catalytic performance in alkene epoxidation with $O_2$. Interestingly, the turnover frequency and mass-specific activity are significantly enhanced by 10 times and 30 times with increasing Co loading from 5.4 wt% to 21.2 wt% in trans-stilbene epoxidation, respectively. Further theoretical studies reveal that the electronic structure of densely populated Co atoms is altered through charge redistribution, resulting in less Bader charger and higher d-band center, which are demonstrated to be more beneficial for the activation of $O_2$ and trans-stilbene. The present study demonstrates a new finding about the site interaction in densely populated single-atom catalysts, shedding insight on how density affects the electronic structure and catalytic performance for alkene epoxidation.

Single-atom catalysts (SACs) have become one of the most attractive frontier research fields in heterogeneous catalysis, with features such as 100% atom utilization, unique electronic and unsaturated coordination structure[1–5]. Under most circumstances, metal single atoms were anchored by heteroatoms (N/O/P/S, etc.) on supports[6–11]. Thus, the coordination microenvironment of central atoms, including the types of coordination atoms, coordination numbers, and even peripheral coordination atoms in the second or higher shell, is crucial for determining their electronic and geometric structures, which in turn influence the absorption or desorption of reaction species and thus the catalytic properties of SACs[12–16]. The overwhelming of SACs research has been focused on this field, with remarkable achievements obtained

in recent decades[17–19]. However, how to further tailor the electronic structure beyond the aforementioned strategy to enhance catalytic performance becomes a new frontier for SACs studies.

An emerging class of densely populated SACs with unique geometric and electronic structures has recently been reported from both experimental and theoretical asepcts[20–28]. An apparent advantage of densely populated SACs is the higher mass-specific activity, which is significant in maximizing reactor productivity in large-scale industries[22,24,29]. Another potential advantage of densely populated SACs is the possibility of an additional type of interaction among single sites, which can further influence the local geometric or electronic structure of individual metal centers via electron transfer, spin

[1]Beijing National Laboratory for Molecular Sciences, CAS Research/Education Center for Excellence in Molecular Sciences, Laboratory of Molecular Nanostructures and Nanotechnology, Institute of Chemistry, Chinese Academy of Sciences, 100190 Beijing, PR China. [2]School of Chemical Sciences, University of Chinese Academy of Sciences, 100049 Beijing, PR China. [3]Department of Materials Science and Engineering, Beijing Jiaotong University, 100044 Beijing, PR China. [4]Key Laboratory of Soil Environment and Pollution Remediation, Institute of Soil Science, Chinese Academy of, 210008 Nanjing, PR China. [5]These authors contributed equally: Hongqiang Jin, Kaixin Zhou. ✉e-mail: yuyu@bjtu.edu.cn; pxcui@issas.ac.cn; cycao@iccas.ac.cn

coupling or charge redistribution, and thus affect the intrinsic activity of active sites[15,20,21,23,30]. For example, Zeng et al.[23] demonstrated that the synergetic interaction between neighboring Pt atoms on $MoS_2$ induced distinct reaction paths and improved the catalytic activity in $CO_2$ hydrogenation. Lu et al.[22] developed a two-step annealing method to synthesize a series of Cu SACs with varying densities and observed that the catalytic activity in the azide-alkyne cycloaddition reaction was proportional to Cu single-atom density. Wang et al.[31] recently reported a strategy for modulating the site distance by varying the atomic Cu density, and discovered that Cu SAC with moderate density exhibited the highest activity for peroxy-disulfate activation in Fenton-like reactions.

These examples indicated that the interactions among single sites in densely populated SACs could indeed synergize to further regulate the electronic structure and reactivity in various reactions[20,31,32]. However, the underlying mechanisms were distinct, and there was no unified theoretical guidance available at the same time. Therefore, much more effort is needed to investigate such an interaction in various reactions, which will not only give a deeper mechanistic understanding of structure-performance relationship at the atomic scale, but will also provide guidance for developing more efficient SACs for catalytic reactions.

Herein, we report a general and reliable strategy to synthesize various densely populated M-SACs (M = Fe, Co, Ni, Cu, Zn, Ru, and Ir) with loading up to 35.5 wt%. Taking cobalt as an example, we further produced a series of Co SACs with loadings ranging from 5.4 wt% to 21.2 wt% to investigate the site interaction effect on regulating the electronic structures of Co atoms and their catalytic performance. Aberration-corrected high-angle annular dark-field scanning transmission electron microscopy (AC HAADF-STEM) and X-ray absorption fine structure (XAFS) characterizations verified that all of the Co SACs were atomically dispersed with the same coordination structures. However, with loadings increase, the valence states of Co species shift to metallic, indicating that the electronic structures of Co single atoms are changed, which can be attributed to the interactions among single sites in densely packed SACs through charger redistribution. In the trans-stilbene epoxidation with $O_2$, the turnover frequency and mass-specific activity were significantly enhanced by 10 times and 30 times with increasing Co loadings from 5.4 wt% to 21.2 wt%, respectively. Further experimental and theoretical studies revealed that the electronic structure of Co atoms in densely populated Co SACs resulted in less Bader charger and higher d-band center, which were more beneficial for the activation of $O_2$ and trans-stilbene.

## Results

### General synthesis and characterizations of densely populated metal SACs

A two-step strategy incorporating polycondensation and subsequent pyrolysis procedures was developed to synthesize the densely populated metal SACs (Supplementary Fig. 1, details are shown in the Experimental Section)[32]. The key success of this strategy relies on the controllable polycondensation during the first step, whereby the small molecules (melamine, cyanuric acid, L-alanine and phytic acid) are spontaneously polymerized in water to form two-dimensional nanosheets (Supplementary Fig. 2). At the same time, metal ions are complexed in it by the surrounding heteroatoms (N/O). Metal SACs with N/O-coordination structure are produced after pyrolysis at a moderate temperature (700 °C) in an Ar atmosphere. Because the precursor contains abundant sites for anchoring metal ions, densely populated metal SACs with loadings up to 35.5 wt% can be achieved.

As shown in Fig. 1a, seven different types of densely populated metal SACs were produced. In principle, this efficient synthetic approach can be extended to produce other metal SACs. According to inductively coupled plasma atomic emission spectroscopy (ICP-AES) results, the metal loadings were all higher than 10 wt% (Supplementary

Table 1). X-ray diffraction (XRD) patterns show only a sole broad feature at ~22°, corresponding to the interlayer distance of the carbon matrix, excluding the existence of any crystalline species in these densely populated metal SACs (Supplementary Fig. 3). Scanning electron microscopy (SEM), transmission electron microscopy (TEM), and corresponding energy-dispersive X-ray spectroscopy (EDS) mapping images demonstrate that all of these metal species are highly dispersed throughout the whole two-dimensional laminar structured carbon matrix doped with heteroatoms (Supplementary Figs. 4–9).

The atomic dispersion of metal species was then directly visualized by employing AC HAADF-STEM. Due to the heavy Z-contrast, the bright single dots ascribing to metal single atoms can be clearly seen in their typical images (Fig. 1b–g). Moreover, Fourier-transformed $k^3$-weighted extended XAFS (FT-EXAFS) spectra show only one prominent peak at -1.6 Å (without phase shift) corresponding to metal-N/O scattering path and the absence of metal-metal scattering, further confirming the atomically dispersed metal atoms in all SACs samples (Fig. 1h–m). Wavelet transform (WT) images also show only one intensity of M-N/O-coordination scattering (Supplementary Fig. 10). Further quantitative EXAFS fitting results suggested an average coordination number (CN) of -4 for all metal SACs (Supplementary Table 2). Because N and O atoms are difficult to differentiate in EXAFS, X-ray photoelectron spectroscopy (XPS) was employed to investigate the metal-N/O bonds. N $1s$ and O $1s$ XPS spectra show the existence of metal-N and metal-O peaks (Supplementary Figs. 11 and 12). These results suggested metal single atoms were dual-coordinated with N and O atoms in these SACs, within a total CN of four. For convenience, these densely populated metal SACs were denoted as $M_1$/NOC, M represents metal.

### Controllable synthesis and characterizations of Co SACs with various densities

Metal SACs with different densities can also be easily obtained by adjusting the amounts of metal precursor based on the proposed strategy, which provides a reliable way to investigate the site interaction effect in densely populated SACs. Taking Co as an example, three Co SACs with loadings of 5.4, 10.9, and 21.2 wt% were prepared (Supplementary Table 3). The corresponding samples were designated as $Co_1$/NOC-5, $Co_1$/NOC-11 and $Co_1$/NOC-21, respectively. SEM images demonstrate the similar morphology of these $Co_1$/NOC samples (Supplementary Fig. 13). XRD, TEM and corresponding EDS mapping characterizations confirmed highly dispersed Co species in these Co SACs samples (Supplementary Figs. 14–17). As shown in Fig. 2a–c, AC HAADF-STEM images present that Co species are all atomically dispersed, and the density of Co single atoms becomes higher from $Co_1$/NOC-5 to $Co_1$/NOC-21. This would probably induce a lot of adjacent single atoms and result in different electronic structure in densely populated Co SACs.

XAFS measurements were then conducted to investigate the electronic properties and coordination structures of these Co SACs with different loadings. Figure 2d shows the normalized Co K-edge XANES spectra. It can be seen that the absorption threshold positions were located between CoO and Co foil, suggesting that the valences of Co species were all between 0 and +2. However, the position of the rising edge exhibited a downshift from $Co_1$/NOC-5 to $Co_1$/NOC-21, indicating that the Co oxidation state became lower with increasing Co density. Co $2p$ XPS spectra also exhibited a similar tendency (Supplementary Fig. 18). These differences suggest the electronic structures of central Co atoms are different. The corresponding FT-EXAFS spectra exhibits only a prominent peak at -1.52 Å of Co-N/O and without Co−Co peak at -2.2 Å (Fig. 2e), confirming the atomically dispersed metal atoms in $Co_1$/NOC-x samples. WT images also show only one maximum intensity at -5 Å$^{-1}$ of Co-N/O scattering path (Fig. 2f). Moreover, a small k-value up-shift can be observed at the WT contour plots of samples compared to CoPc. This can be ascribed to the introduction of O in the first-coordination shell of Co atoms, implying that Co single atoms

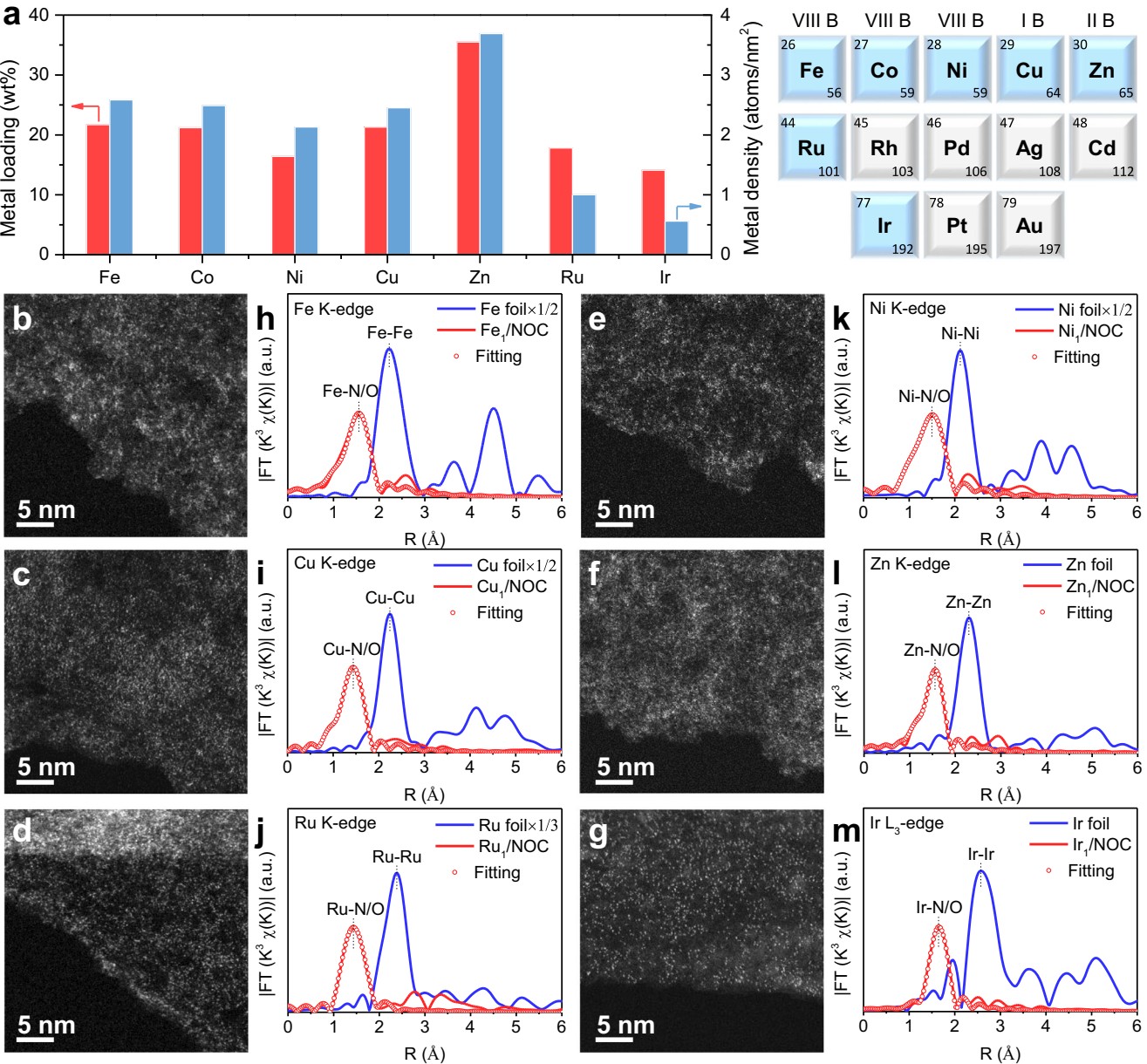

**Fig. 1 | Synthesis and characterizations of M-SACs. a** Metal loadings in M-SACs and schematic diagram of prepared M-SACs. **b–g** AC HAADF-STEM images and (**h–m**) $k^3$-weighted Fourier transform spectra of (**b, h**) Fe$_1$/NOC, (**e, k**) Ni$_1$/NOC, (**c, i**) Cu$_1$/NOC, (**f, l**) Zn$_1$/NOC, (**d, j**) Ru$_1$/NOC, and (**g, m**) Ir$_1$/NOC.

were coordinated with N/O dual-atoms. Meanwhile, the first-coordination shell of Co-P was excluded (Supplementary Fig. 19). Quantitative EXAFS curve-fitting results show the average CNs of Co single atoms in all SACs are estimated to be 3.8–4.2 (Supplementary Table 4). Element analysis shows that the ratios of N to O are also kept as ~3 in Co$_1$/NOC-x (Supplementary Table 5). Furthermore, the Raman and C 1s XPS spectra of Co$_1$/NOC-x demonstrate a similar local structure of supports (Supplementary Fig. 20). These results suggest that these Co SACs might maintain the identical Co$_1$-N$_3$O$_1$ coordination structure with varying site density. That is to say, the shift of Co valence states and different electronic structures of central Co atoms should be attributed to the different site density in these Co SACs, rather than coordination structure or coordination numbers.

## Theoretical understanding the difference of Co SACs with various densities

In order to better understand the influence of site density on the electronic structures of Co single atoms from theoretical, different

numbers of Co$_1$-N$_3$O$_1$ single sites were evenly embedded in the same 6*6 cell of N/O doped graphene to represent Co SACs with different densities (Supplementary Fig. 21). The maximum of four Co$_1$-N$_3$O$_1$ sites can be accommodated in such a grid area. Under this situation, the corresponding theoretical mass loading of Co atom was calculated as 22 wt%, which was quite close to the actual loading of as-synthesized Co$_1$/NOC-21 sample (Supplementary Fig. 21a). Especially, when the number of Co$_1$-N$_3$O$_1$ single sites were set to one and two, respectively, the theoretical Co loadings were calculated to be 6 wt% and 12 wt%, which also agreed well with the experimental results of Co$_1$/NOC-5 and Co$_1$/NOC-11 samples (Supplementary Fig. 21b, c). Thus, we have effectively correlated the Co metal loading and Co single-atom density, which is highly useful in guiding us in the construction of efficient SACs. These models with different numbers of Co$_1$-N$_3$O$_1$ single sites are reliable to represent Co SACs with different densities for theoretical investigation. DFT calculation results also indicated the high stability of these configurations after optimization, with very negative adsorption energies in theory (Supplementary Fig. 22). Moreover,

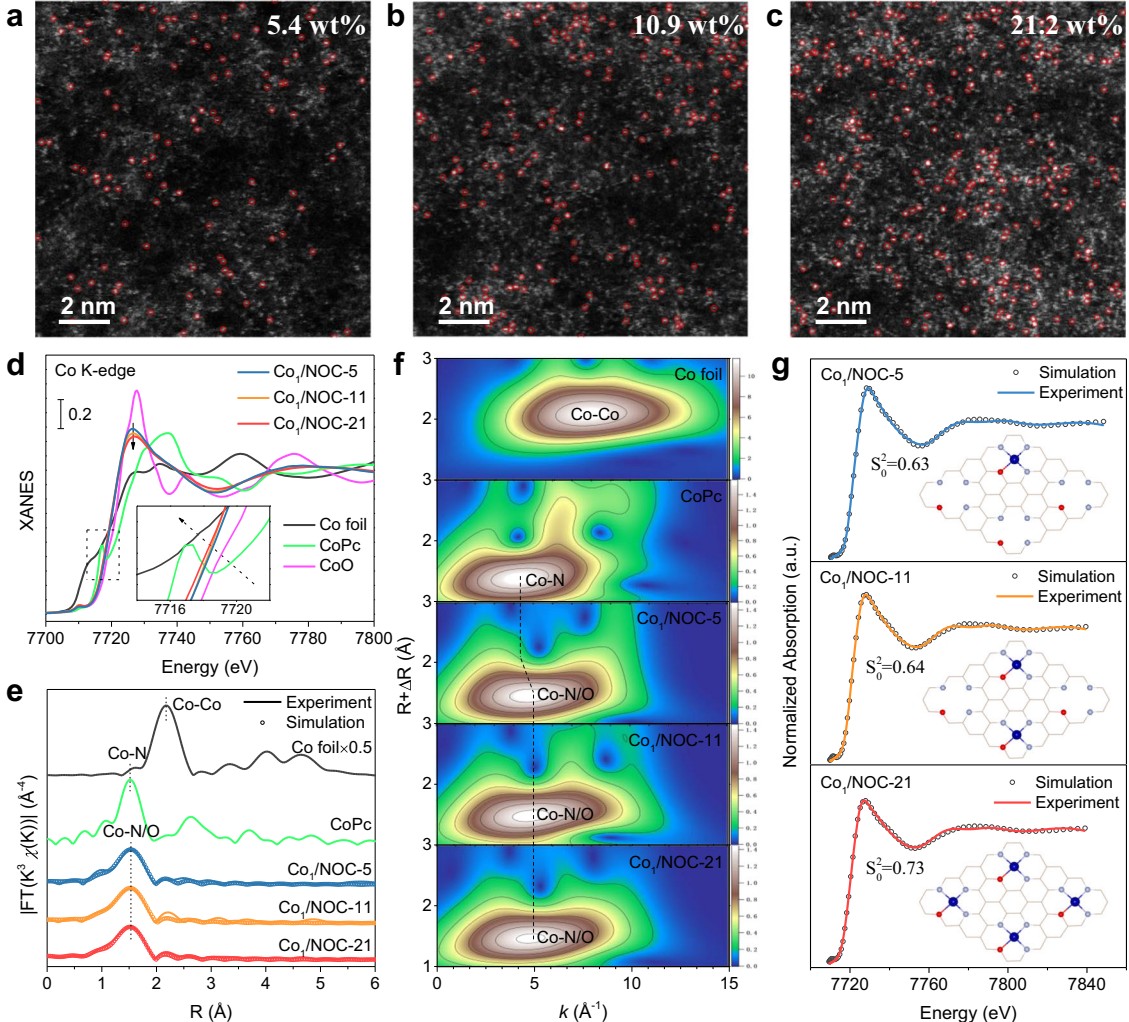

**Fig. 2 | Characterizations and atomic structural analysis of Co SACs with different density. a–c** AC HAADF-STEM images of 5.4 wt%, 10.9 wt%, and 21.2 wt% Co SAC samples. Co single atoms are marked in red circles. **d** Normalized Co K-edge XANES spectra. **e** $k^3$-weighted Fourier transform spectra of $Co_1/NOC$-x and reference samples. **f** Corresponding wavelet transform (WT) analysis. **g** The Co K-edge XANES experimental spectra (solid lines) and the theoretical spectra (dots) calculated with the depicted structures (insert) of $Co_1/NOC$-x samples.

XANES simulation curves with these models matched well with $Co_1$-$N_3O_1$ coordination configuration (Fig. 2g), further confirming the possibility of $Co_1$-$N_3O_1$ coordination structure in all Co SACs.

According to the optimized models, the charge density differences of Co single sites were then calculated. As shown in Fig. 3a, it can be seen that charge was transferred from Co atoms to the supported matrix. With increasing number of $Co_1$-$N_3O_1$ sites from 1 to 4, the whole support became more charge connections, which resulted in the charge redistribution. Bader charge analysis further suggested that the average charge transfer of Co atom became gradually less from 4-$Co_1$-$N_3O_1$ to 1-$Co_1$-$N_3O_1$ (Supplementary Table 6), which was consistent with the XAFS and XPS results. These differences in charge density confirmed that the electronic structures of Co atoms can be altered through charge redistribution in densely populated Co SACs. Further projected density of states (pDOS) calculations results showed that the Co d-band center of 4-$Co_1$-$N_3O_1$ was up-shifted and gradually closed to the Fermi level ($E_F$) in comparison with 1-$Co_1$-$N_3O_1$, in consistent with Bader charge analysis results (Fig. 3b, c).

In addition, as shown in Supplementary Fig. 23, because of the asymmetry from the spin-up to spin-down channel, the ground state of 1-$Co_1$-$N_3O_1$ was calculated to be ferromagnetic with a total spin moment of 0.797 $\mu_B$, indicating a weak interaction among $Co_1$-$N_3O_1$ moieties in low-density Co SAC. On the contrary, because of the

slightly symmetrical distribution in spin-up and spin-down channel, the ground state of 4-$Co_1$-$N_3O_1$ exhibits a reduced spin moment of 0.310 $\mu_B$ equally distributed on all Co atoms, suggesting very strong interaction between adjacent $Co_1$-$N_3O_1$ moieties in high-density Co SAC (Supplementary Table 7). The decrease of spin moment from 1-$Co_1$-$N_3O_1$ to 4-$Co_1$-$N_3O_1$ can be ascribed to the downshift of energy of the Co 3d orbital as adjacent Co atoms get closer (Fig. 3d), which was consistent with the reported results in literatures[21,33]. To sum up, from both experimental and theoretical analysis, we verified that the interaction among these single sites in densely populated Co SACs could indeed alter the charge density and electronic structure of Co atoms through charge redistribution, which would further affect their catalytic performance.

## Catalytic performances of Co SACs with various densities for trans-stilbene epoxidation

Epoxides are of importance in fine chemical industry and organic synthesis[34–36]. In current processes for alkene epoxidation, a large number of expensive oxidants or co-reagents are usually required, which inevitably leads to high cost[37,38]. To address this problem, several SACs (Fe, Ag, Pt, Pd, etc) had been developed and exhibited excellent catalytic performances, allowing $O_2$ to be the oxidant without any co-reagents[29,37,39–41]. Inspired by these findings, we explored the potential

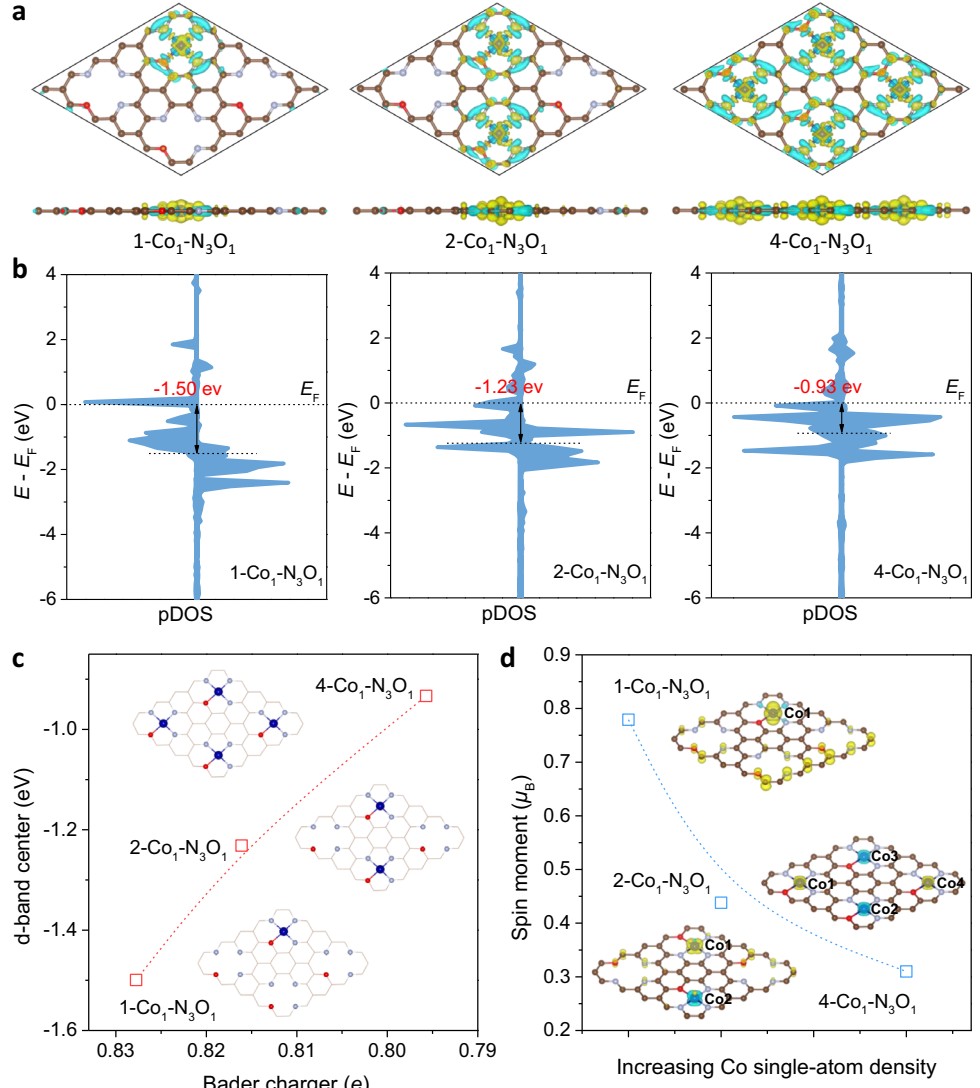

**Fig. 3 | Theoretical analysis of electronic structure of Co1/NOC with different densities. a** Top and side views of differential charge density of $x$-$Co_1$-$N_3O_1$ models. Isosurface: $0.005\,e/Å^{-3}$. Color legend for isosurface: blue, charge depletion; yellow, charge accumulation. **b** Projected density of state (pDOS) for the Co $3d$ orbital.

**c** The relationship between Bader charger and d-band center in various $x$-$Co_1$-$N_3O_1$ models. **d** The plot of spin moment with respect to Co single-atom density. Inset: The spin density isosurfaces of $0.005\,e/Å^{-3}$.

application of as-synthsized $M_1$/NOC samples in trans-stilbene (SB) epoxidation under 1 atm $O_2$. First, we investigated the performance of various $M_1$/NOC samples. $Co_1$/NOC SAC can convert up to 96.5% of SB after 1 h, but other transition metal SACs can only convert up to 54.4%, showing that Co SACs have the highest catalytic performance in SB epoxidation (Fig. 4a). Therefore, we focused on the Co SACs samples and disclosed the influence of density on the catalytic property.

As shown in Fig. 4b, there was no reaction detected with P-doped support alone, indicating that Co atoms were the active species in all samples. However, notable differences in the catalytic activity of Co SACs with different density can be observed under the same reaction conditions. Morevoer, the calculated turnover frequency (TOF) was significantly enhanced by 10 times with increasing Co density from $Co_1$/NOC-5 to $Co_1$/NOC-21, while the selectivity always remained at a satisfactory level (>98%), suggesting the outstanding active sites in high-density $Co_1$/NOC-21 (Fig. 4c). Further apparent activation energies ($E_a$) tests show that $Co_1$/NOC-21 has an $E_a$ of 58.2 kJ mol$^{-1}$, which is much lower than that of $Co_1$/NOC-11 and $Co_1$/NOC-5, confirming the excellent catalytic performance of densely populated Co SACs (Supplementary Fig. 24). Particularly, the mass-specific activity is more

important for industrial applications. Due to the high metal loading and intrinsic activity, the mass-specific activity of $Co_1$/NOC-21 reached as high as 193 mol·g$^{-1}$·h$^{-1}$, which was nearly 30 times and even 15 times to that of $Co_1$/NOC-5 and Co nanoparticles (Co NPs), respectively. Such high mass-specific activity was also much higher than the values reported in literatures[37,40]. Moreover, $Co_1$/NOC-21 can tolerate a broad scope of substrates in alkene epoxidation (Supplementary Table 8) and exhibits outstanding catalytic stability after seven consecutive cycles (Supplementary Fig. 25). The microstructure and electronic structure of Co single atoms after re-used also maintained nearly the same and no leaching of Co species was observed (Supplementary Figs. 26 and 27 and Supplementary Table 3).

Because of the facile synthesis strategy and excellent catalytic performance, we can readily achieve gram-scale production of densely populated Co SACs for SB epoxidation (Fig. 4d). XAFS measurement demonstrated the atomically dispersed state of gram-scale Co SAC with no appreciable difference compared with $Co_1$/NOC-21 (Supplementary Fig. 28). Under optimized reaction conditions, the gram-scale Co SAC was evaluated to be effective for SB epoxidation with a high yield of 92.6% (Fig. 4e). The obtained SB epoxidation product (SBO)

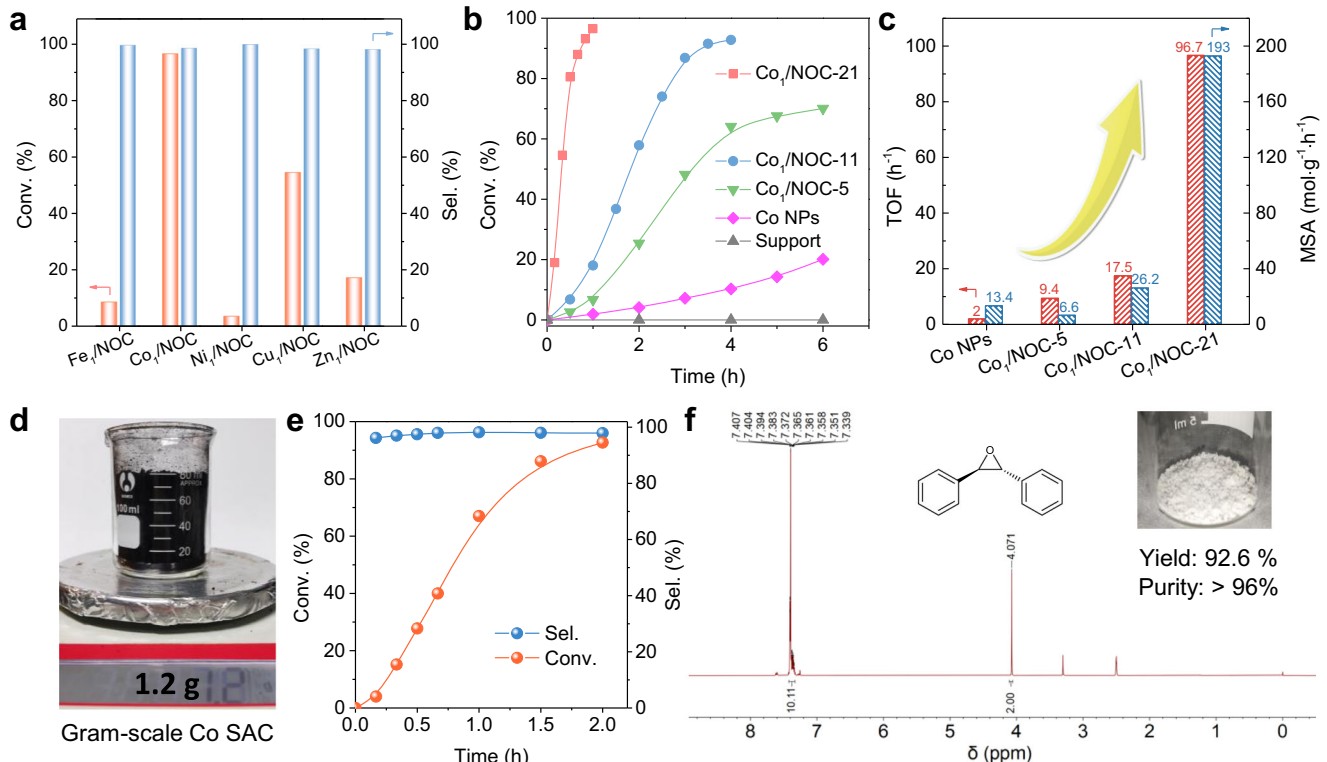

**Fig. 4 | Catalytic performances in trans-stilbene epoxidation. a** The catalytic performance of $M_1/NOC$ samples in trans-stilbene epoxidation within 1 h. **b** The dynamics plots of trans-stilbene conversion against reaction time over various samples. Reaction condition: 1 mmol trans-stilbene epoxidation, containing 0.01 mmol Co in each catalyst (10 mg support), 10 mL solvent, 140 °C, $O_2$ balloon. **c** TOF and mass-specific activity (MSA) of various $Co_1/NOC$ samples. **d** Optical photo of as-prepared gram-scale Co SAC sample. **e** The dynamics plots of SB conversion against reaction time over gram-scale Co SAC sample. **f** $^1H$ NMR spectra of generated SBO. (400 MHz, $D_6$-DMSO) δ 7.34−7.40 (m, 10H), 4.07 (s, 2H). Inset: molecular structure and optical photo of product SBO.

was characterized by nuclear magnetic resonance (NMR, Fig. 4f). In the $^1H$ NMR spectrum, the triplet peak at 4.07 ppm corresponds to the hydrogen adjacent to the epoxy group, and the multiple peak with a chemical shift at 7.34−7.40 ppm corresponds to the aromatic hydrogen, indicating the high purity of SBO (>96%) compared with commercial production (Supplementary Fig. 29). The peak splitting and peak area coupled with $^{13}C$ NMR were all consistent with the SBO (Supplementary Fig. 30). Overall, the densely populated Co SAC displayed superior catalytic performance, showing the potential for industrial application in stilbene epoxidation.

## Theoretical understanding the density effect on the catalytic performance

As shown in above experimental results, Co SACs with different densities exhibited different catalytic performance for SB epoxidation. To reveal the reaction mechanism and the influence of electronic structures of Co single atoms induced by the density on the catalytic performance, DFT calculations were further performed. We chose $1-Co_1-N_3O_1$ and $4-Co_1-N_3O_1$ to represent the lowest density and highest density Co SACs, respectively. From the kinetic curves, it can be found that SB epoxidation was the consecutive reaction, implying the activation of $O_2$ was the first step. Moreover, the calculated adsorption energies suggested that Co sites in $x-Co_1-N_3O_1$ were more favorable for the $O_2$ adsorption rather than SB molecule (Fig. 5a). In addition, we investigated the possibility of $O_2$ adsorption on a defective $N_3O_1$ vacancy. The calculated O−O bond distance after adsorption on vacancy was only 1.25 Å (equal to that of free $O_2$), implying $O_2$ molecules can not be activated on defective $N_3O_1$ vacancies (Supplementary Fig. 31). Therefore, we focused on the electronic structure of Co atoms and $O_2$ molecules to understand the activation mechanism.

As shown in Fig. 5b, we can find that it presents a positive correlation between the catalytic activity and average Bader charge of Co atoms, where a sharp enhancement of TOF value realizes with the decrease of Bader charge, suggesting that the Bader charge might be a key descriptor in trans-stilbene epoxidation. Therefore, we investigated the charge density difference and Bader charge transfer between Co atom and $O_2$ (Fig. 5c), which revealed that $^*O_2$ gained 0.36 $e$, 0.38 $e$ and 0.44 $e$ from Co atom in $1-Co_1-N_3O_1$, $2-Co_1-N_3O_1$, and $4-Co_1-N_3O_1$, respectively, suggesting more electrons were filled into the $O_2$ $2\pi^*$ orbital to active $O_2$ on $4-Co_1-N_3O_1$ (Supplementary Fig. 32)[42]. Compared with O−O bond length of 1.29 Å on $1-Co_1-N_3O_1$, it was enlarged to 1.31 Å on $4-Co_1-N_3O_1$, indicating Co single atoms with higher charge density in $4-Co_1-N_3O_1$ was more favorable for activation of $O_2$ molecule. Furthermore, we investigated the spin moments of Co atoms after $O_2$ adsorption (Supplementary Fig. 33 and Supplementary Table 9). It was found that there was more pDOS overlap and a lager energy splitting between bonding and antibonding orbitals (Supplementary Fig. 34). $4-Co_1-N_3O_1$ showed the largest spin moment of 0.443 $\mu_B$. Meanwhile, the adsorbed $O_2$ also exhibited the largest atomic or molecular spin moments, which was consistent with previous work[43]. The larger variations in spin moment of $O_2$ in $4-Co_1-N_3O_1$ before and after oxygen adsorption demonstrates the most intensive electron transfer from Co to $O_2$ (Supplementary Table 10). As a consequence, when $O_2$ was adsorbed on high-density Co with the lowest activation energy, the biggest increase in the O−O bond distance occurred.

The reaction pathways and the corresponding energy profiles over these three $Co_1-N_3O_1$ models were finally investigated using DFT calculations (Fig. 5d and Supplementary Figs. 35−37). The $O_2$ molecules were firstly covered on all Co single atoms, with exothermic values of −1.10, −1.12 and −1.15 eV over 1, 2, $4-Co_1-N_3O_1$ models, respectively (II). Subsequently, an SB molecule attacked

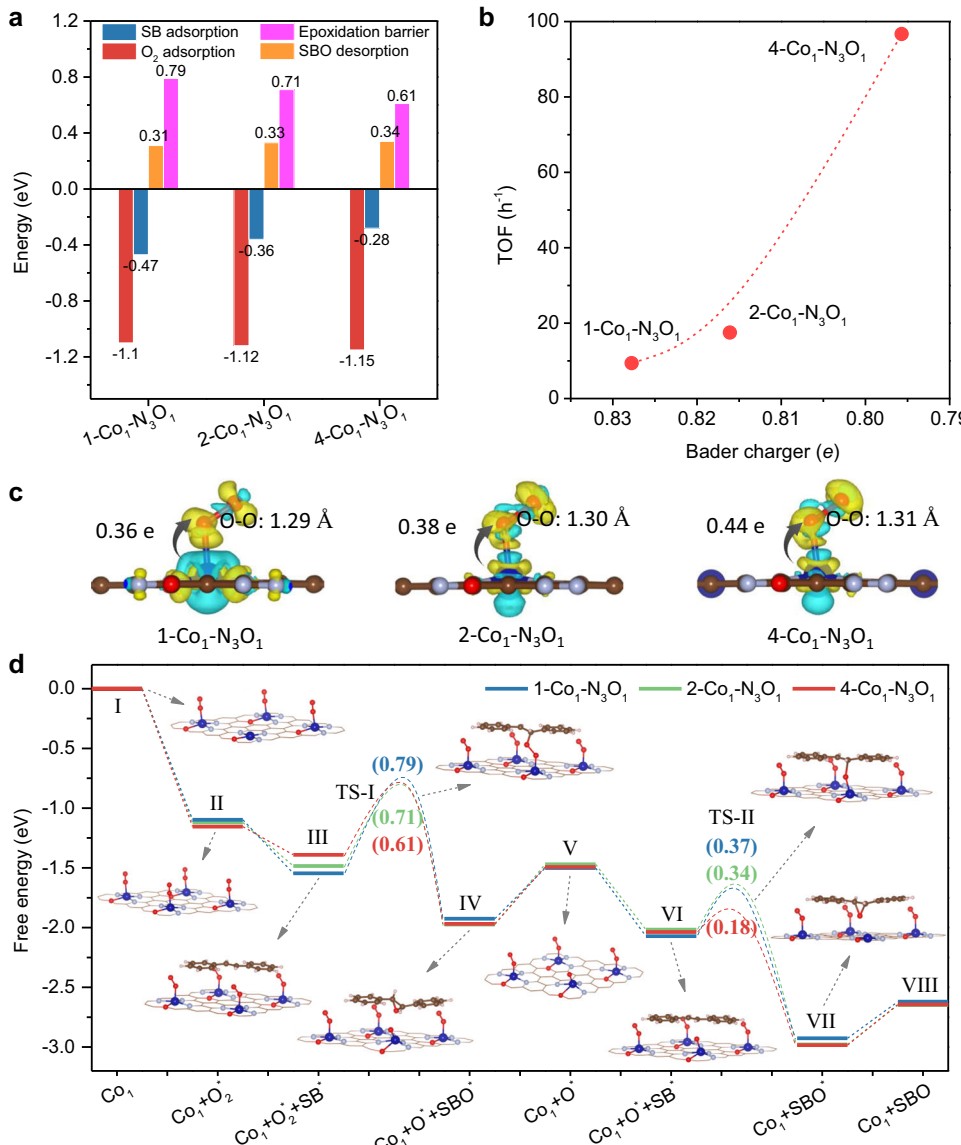

**Fig. 5 | Theoretical calculation of adsorption configurations and reaction pathways. a** The calculated energies for adsorption $O_2$ and SB, as well as SBO desorption energies. **b** The relationship between Bader charge and catalytic activity. **c** Charge density differences after $O_2$ adsorption on various x-$Co_1$-$N_3O_1$ models. **d** Energy profiles of trans-stilbene epoxidation reaction over 1-$Co_1$-$N_3O_1$, 2-$Co_1$-$N_3O_1$, and 4-$Co_1$-$N_3O_1$ models. Inset: the configurations of intermediates of 4-$Co_1$-$N_3O_1$.

one $O_2$ molecule pre-adsorbed $Co_1$ site (III) for activation and transformed to SBO with energy barriers (TS-I) of 0.79, 0.71 and 0.61 eV on 1, 2, 4-$Co_1$-$N_3O_1$, respectively (Supplementary Table 11). The residual oxygen can be converted to a new one-coordinated oxygen atom by overcoming relative low-energy barriers (IV to V) and then participated in the next steps for SB epoxidation reaction (VI to VIII). Similarly, the energy barrier (TS-II) on 4-$Co_1$-$N_3O_1$ was also lower than that on 1-$Co_1$-$N_3O_1$ and 2-$Co_1$-$N_3O_1$. Upon the desorption of the second SBO molecule, as well as the adsorption of another $O_2$ molecule, the catalytic reaction cycle started again. From the whole profile, it can be seen that Co single atom in 4-$Co_1$-$N_3O_1$ with higher charge density was more favorable for the activation of $O_2$ and SB substrates (Supplementary Table 12).

Moreover, we also calculated the energy changes of the reaction over 4-$Co_1$-$N_3O_1$ model in which the two benzene rings interact with two adjacent Co sites simultaneously (Supplementary Fig. 38). The energy barrier for the oxidation of the benzene ring by two adjacent Co sites was calculated to be 2.02 eV (formation of epoxy benzene) or

2.60 eV (formation of phenol), which was much higher than the corresponding value of the SB epoxidation path (0.61 eV, Fig. 5d). Therefore, the interaction of SB with two adjacent Co sites can be excluded as the main reaction path in SB epoxidation.

Finally, we used DFT calculations to understand the trans-stilbene epoxidation reaction pathways of Co NPs. Both XRD and HRTEM characterizations revealed that Co NPs exposed with Co(111) crystal faces (Supplementary Figs. 39 and 40). Therefore, we used Co(111) to represent Co NPs/NC sample for calculations. As shown in Supplementary Fig. 41, an $O_2$ molecule was adsorbed and dissociated to O* on Co(111) with a high exothermic of −4.56 eV (III). The SB molecule was then activated and converted to SBO with an energy barrier of 1.21 eV (IV−V). By overcoming another energy barrier of 1.36 eV (V to VI), the remaining oxygen can be transformed to a new one-coordinated oxygen atom and subsequently participate in the next step of the SB epoxidation cycle. The highest energy barrier for SBO production reached to 1.78 eV (VII to VIII). It can be seen that the energy barrier on Co NPs was much higher than on $Co_1$/NOC-x samples, resulting in the

lowest reaction rate in trans-stilbene epoxidation, which was consistent with experimental results.

## Discussion

In summary, we developed a versatile strategy to synthesize various densely populated metal SACs, and then produced a series of Co SACs with loadings ranging from 5.4 wt% to 21.2 wt% for demonstration to investigate the site interaction effect on regulating the electronic structures of Co atoms and their catalytic performances. Taking trans-stilbene epoxidation with $O_2$ as a model reaction, we found that the reaction rate and mass-specific activity of Co SAC with high loading of 21.2 wt% were 10 times and 30 times, respectively, compared to 5.4 wt% Co SAC. Further experimental and theoretical studies revealed that the electronic structure of Co atoms in densely populated Co SACs was altered through charge redistribution, resulting in less Bader charger and higher d-band center, which were demonstrated to be more beneficial for activation of $O_2$ and trans-stilbene. The findings in this work would deepen the mechanistic understanding of the structure-performance relationship at the atomic scale in densely populated SACs and provide guidance for designing more efficient SACs for catalytic reactions.

## Methods

### Synthesis of Co₁/NOC-x (where x is the weight fraction of Co)

To obtain the Co₁/NOC-21 sample, it was prepared according to our previously reported method with a slight modification. In detail, melamine (2 g), cyanuric acid (2 g), L-alanine (2 g), and phytic acid solution (50 wt%, 200 μL) were added and dispersed in 100 mL of deionized (DI) water. Afterward, the suspension was strongly stirred at 100 °C to get a homogenous polymeric precursors. In the meantime, 270 mg of Co(NO₃)₂·6H₂O were slowly added into the above precursors under stirring for several hours, yielding a black slurry. Then, the resultant slurry was quickly frozen in liquid nitrogen. After freeze drying, the collected powder was pyrolyzed at 700 °C for 2 h under Ar gas to obtain the final sample without further treatment. For the synthesis of Co₁/NOC-5 and Co₁/NOC-11 sample, the procedure is the same as that of Co₁/NOC-21, where only the weight of Co(NO₃)₂·6H₂O was changed to 70 mg and 150 mg, respectively. For the synthesis of Co NPs/NC sample, the as-prepared Co₁/NOC-21 was transferred into a tube furnace maintaining 500 °C for 2 h under a flowing mixture of 5% H₂/Ar atmosphere with a heating rate of 5 °C min⁻¹. For the synthesis of the gram-scale Co SAC sample, the procedure is the same as that of Co₁/NOC-21, except that the weight of all the raw materials is amplified by five times.

### Synthesis of M-SACs (M = Fe, Ni, Cu, Zn, Ru, Ir)

M-SACs samples were synthesized using similar procedures of Co₁/NOC-x, except that the amounts of metal salt were adjusted. In detail, Fe(NO₃)₃·9H₂O (400 mg), Ni(NO₃)₂·6H₂O (270 mg), Cu(NO₃)₂·4H₂O (210 mg), Zn(NO₃)₂·6H₂O (350 mg), RuCl₃·3H₂O (70 mg), and IrCl₃·xH₂O (30 mg) were used as the corresponding metal precursors.

### Material characterizations

The morphologies and structures of the samples were performed on transmission electron microscopy (TEM) (JEM-2100F, JEOL, Japan) and scanning electron microscopy (SEM) (HITACHI S-4800, Japan). Element mapping was recorded on TEM equipped with Oxford detection. The powder X-ray diffraction (XRD) patterns were characterized on a Rigaku D/max-2500n diffractometer with Cu Kα radiation ($\lambda = 1.5418$ Å) at 40 kV and 200 mA. X-ray photoelectron spectroscopy (XPS) measurements were measured on a VG Scientific ESCALab220i-XL electron spectrometer using 300 W Al kα radiation. Inductively coupled plasma atomic emission spectroscopy (ICP-AES) was employed with Shimadzu ICPE-9000 to confirm the loading content of metal on the catalysts. The AC HAADF-STEM images were performed on JEOL ARM300F at 300 kV, equipped with a probe spherical

aberration corrector. ¹H and ¹³C NMR spectra were acquired via a Bruker Advance III HD-400 MHz spectrometer with a BFO smart probe.

### XAS measurements and analysis

The cobalt K-edges XAFS spectra of the standards and samples were collected at the beamline 1W1B of the Beijing Synchrotron Radiation Facility (BSRF). The typical energy of the storage ring was 2.5 GeV and the electron current was ~250 mA in the top-up mode. The white light was monochromatized by a Si (111) double-crystal monochromator and calibrated with a Co foil (K-edge at 7709 eV). Samples were pressed into thin slices, and positioned at 45° to the incident beam in the sample holder. The XAFS spectra were recorded in fluorescence mode with a Lytle detector oriented at 90° to the incoming beam.

The XAFS data were analyzed using the software packages Demeter. The spectra were normalized using Athena firstly, and then shell fittings were performed with Artemis. The $\chi(k)$ function was Fourier-transformed (FT) using $k^3$ weighting, and all fittings were done in R-space. The coordination parameters of samples were obtained by fitting the experimental peaks with theoretical amplitude. The quantitative curve-fittings were conducted with a Fourier transform k-space range of 2.7–11.8 Å⁻¹. The backscattering amplitude F(k) and phase shift Φ(k) were calculated by FEFF7.0 code. While the curve-fitting, all the amplitude reduction factor $S_0^2$ was set to the best-fit value of 0.769 determined from fitting the data of copper foil by fixing coordination numbers as the known crystallographic value. The wavelet transformed (WT) $\chi(k)$ function of samples were performed using the Igor pro script developed by Funke et al.[44]. The Morlet wavelet was chosen as the basis mother wavelet, and the parameters ($\eta = 6$, $\sigma = 1$) were used for a better resolution in the wave vector $k$.

The quantitative XANES calculation was analyzed using the MXAN package. The XANES spectrum from the absorption edge up to 120 eV were calculated via comparison between experimental data and theoretical calculations obtained by changing relevant geometrical parameters around the photon absorbed site[45]. The X-ray absorption cross-sections were calculated using the full multiple scattering approaches in the framework of the muffin-tin (MT) approximation for the shape of the potential[46]. In this case, the calculation clusters include ~70 atoms within a radius of 7.0 Å.

### Theoretical calculations

The calculations were all based on density functional theory using the Vienna ab initio simulation package (VASP)[47]. Generalized gradient approximation (GGA)[48] and Perdew−Burke−Ernzerhof (PBE)[49] exchange-correlation functional in PBE + U mode (U$_{eff}$ = 1 for Co) was used. The van der Waals correction was employed by DFT-D3 method in all calculations. 3d and 4 s valence electrons were considered for Co taking into account of spin polarization when 2 s and 2p for C, N, and O. The cut-off energy, total energy convergence, and force convergence were set as 500 eV, $1 \times 10^{-3}$ eV and 0.01 eV Å⁻¹ respectively. A $6 \times 6$ supercell containing 72 C atoms was adopted for pure graphene, and a vacuum layer of 20 Å was built to get rid of interaction from adjacent cells. Co-N₃O₁ single-atom site was constructed by substituting 1-Co atom for 2 adjacent C atoms, and the 4 neighbors coordinated C atoms were also replaced by 3 N atoms and 1 O atom. N₃O₁ vacancy site was similar besides the Co atom that was absent. In the three models ($x$-Co₁-N₃O₁-gra, $x = 1$, 2, 4), 4-Co₁-N₃O₁-gra possesses four Co-N₃O₁ sites, while 1-Co₁-N₃O₁-gra and 2-Co₁-N₃O₁-gra have 1 Co-N₃O₁ site + 3 N₃O₁ vacancies and 2 Co-N₃O₁ sites + 2 N₃O₁ vacancies, respectively. The Monkhorst−Pack scheme K points grid sampling was set as $5 \times 5 \times 1$ for the irreducible Brillouin zone. For determining the ground state of all Co SACs models, the energies obtained from both non-magnetic and magnetic calculations were examined, including the ferromagnetic and anti-ferromagnetic constructions for magnetic calculations. In the 4-Co₁-N₃O₁-gra model, two spin opposite orientation arrangements for anti-ferromagnetic structures were considered, in which two Co atoms

along the x axis or on the diagonal of unit cell were set as the same spin orientation, while the other two Co atoms were in opposite spin orientation. In the calculations for energy profiles, the models with $O_2$ adsorbed on $N_3O_1$ vacancies (in 1-$Co_1$-$N_3O_1$-gra and 2-$Co_1$-$N_3O_1$-gra) and on adjacent Co sites (in 2-$Co_1$-$N_3O_1$-gra and 4-$Co_1$-$N_3O_1$-gra) were used, as the $O_2$ adsorption on the vacancies and Co sites were revealed as a spontaneous exothermic process, and these $O_2$ adsorbed models were closer to the real states in experiments. The transition states were discovered using climbing image nudged elastic band (CINEB) method, and examined via frequency analysis. The formation energy of $x$-$Co$-$N_3O_1$-gra model was investigated to ensure the stability of single Co atom site imbedding substrate, which was calculated as follows Eq. (1):

$$E_{for} = E(x\text{-}Co\text{-}N_3O_1\text{-}gra) - E(N_3O_1\text{-}gra) - xE(Co\text{-}bulk) \quad (1)$$

where $N_3O_1$-gra represents the anchored-Co-free substrate with $N_3O_1$ vacancies.

### Catalytic performance evaluation

For this epoxidation reaction, trans-stilbene (180 mg, 1 mmol), $Co_1$/NOC-x catalysts (each sample containing 0.01 mmol Co), and N, N-dimethylacetamide (10 mL) were added into a 100-mL flask. Then, an oil pump was employed to remove the air in the flask, and an $O_2$ balloon was used to charge about 1 atm $O_2$. Finally, the reaction system was performed at the desired temperature of 140 °C. The product was collected at the reserved time and immediately analyzed using gas chromatography in combination with mass spectrometry (Shimadzu GCMS-QP2010S). To evaluate the reusability of catalyst, the samples from the last reaction were separated by centrifugation, washing with ethanol, and drying under the vacuum. The turnover frequencies (TOFs) of these $Co_1$/NOC-x samples were calculated according to the experiment results, as following Eq. (2),

$$TOF\left(h^{-1}\right) = \left(1 - \frac{n_t}{n_0}\right) \times \frac{n_0}{n_{Co} \times t} \quad (2)$$

where $n_0$ is the initial mole of trans-stilbene, $n_t$ is the mole of trans-stilbene at $t$ time, $n_{Co}$ is the total mole of Co in the sample, and $t$ is the reaction time in an hour.

### Product purification

The reacted solution was filtered to remove the catalyst, and water was added as a poor solvent to precipitate the crude product. The solution was then filtered to obtain a yellow solid. The yellow solid was dissolved in dichloromethane ($CH_2Cl_2$) and dried over anhydrous sodium sulfate ($Na_2SO_4$) for 24 h. Then $Na_2SO_4$ was filtered off and washed with $CH_2Cl_2$. The filtrate was collected, and the $CH_2Cl_2$ solvent was removed with a rotary evaporator to give a pale yellow solid. Finally, the pale yellow solid was recrystallized from ethanol to obtain the white crystalline product.

## Data availability

The data supporting the findings of this study are available within the article and its Supplementary Information. Source data are provided with this paper. Additional data are available from the corresponding authors on reasonable request.

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

## Acknowledgements

We thank the National Key Research and Development Program of China (Grant No. 2018YFA0208504 and 2018YFA0703503, C.Y.C.), the National Natural Science Foundation of China (NSFC 22272181 C.Y.C., 21932006 W.G.S., 92161112 W.G.S., 51972017 Y.Y., and 42077145 P.X.C.) and the Youth Innovation Promotion Association of CAS (Y2017049, C.Y.C.) for financial support. We thank the beamline 1W1B station in Beijing Synchrotron Radiation Facility (BSRF) and Dr. Lirong Zheng for help in XAFS characterization.

## Author contributions

H.Q.J., C.Y.C., and W.G.S. were responsible for most of the investigations, methodology development, data collection/analysis, and writing the original manuscript. K.X.Z., R.X.Z., and H.J.C. assisted with the data collection and experiment analysis. Y.Y. conducted the DFT calculations. P.X.C. helped to analyze the XAFS results. C.Y.C. was responsible for the funding and resources acquisition, supervising the project, revising and editing the manuscript.

## Competing interests

The authors declare no competing interests.
