## [Peer Review File · Nature Communications]

Regulating the electronic structure through charge redistribution in dense single-atom catalysts for enhanced alkene epoxidationREVIEWER COMMENTS

Reviewer #1 (Remarks to the Author):

Jin et al. reported a combined experimental and computational study of Co single atom catalysts (SACs) showing that the electronic structures in terms of charge distribution can be regulated by the active site density and increasing that density can lead to enhanced catalytic performance in alkene epoxidation. In the computation part, they found less Bader charge and higher d-band center for Co SACs as the density of Co increases. As a result, higher density Co SACs can bind O₂ stronger and activate it to a larger extent. The desorption of the product SBO is also easier. Then the authors explain the experimentally observed enhancement of alkene epoxidation using the computed electronic structure of catalysts and free energy profiles of reactions. Overall, this is a comprehensive and interesting study of the impact of density of SACs on a specific catalytic reaction. However, the significance of this work and the novelty of the mechanistic understanding are not that strong to meet the requirement of publication in Nature Communications. Therefore, I could not recommend its publication in the current form. Specific comments that need to be addressed are as follows:

1. The statement in the abstract as “such an interaction has rarely been studied and its influence on the reaction mechanisms in various reactions remains unclear” (Line 19-20) and another one in the introduction as “(for densely populated SACs) further influence the local geometric or electronic structure of individual metal centers via electron transfer, spin coupling or charge redistribution, and thus affect the intrinsic activity of active sites” (Line 50-52) are not on the same page. As a matter of fact, inter-site interactions and their influence on mechanisms of selected reactions have been studied in a couple of previous work of SACs, including theoretical (ref 20) and experimental studies (ref 21) that the authors have cited, and a few more, like Nano. Lett. 2022, 22, 3744; JACS Au 2021, 1, 2130 (review paper); JPCL 2021, 12, 5233; JPCL 2020, 11, 3962; ACS AMI 2020, 12, 15271; JPCL 2019, 10, 7009. It is true that there is no unified theoretical framework to predict how they influence various reactions, but this work also just focused on one specific type of reaction -- alkene epoxidation. The authors should elaborate the background of their work more accurately.

2. Following the first comment, I am afraid that the novelty of this work is not clearly stated, either in the experimental side, like alternative synthetic strategy of making very high metal loading SACs or much higher yield of alkene epoxidation than other work, or in computational side, like new mechanism that has never been reported before. It is noted that this work is similar to their just published work (ref 28. ACS Catalysis 2023, 13, 1316–1325) of which the title is “Understanding the Density-Dependent Activity of Cu Single-Atom Catalyst in the Benzene Hydroxylation Reaction”. What is the advance provided exclusively in this work?

3. Regarding to the mechanistic understanding, I think the advance reported in this work is probably not impressive as measured by the standard of Nature Communications. My major concern is that the connection between the tuning of electronic structures due to the increase of active site density and the appreciable improvement of experimental yield owing to higher metal loading in SACs is not strong. I did

not see clear evidence to support a hypothesis that the former (in a microscopic level) causes the latter (in a macroscopic level). Since they have calculated the whole free energy profile including activation barrier for all elementary reactions, they may calculate TOF to see how much the rate can be altered due to the increase of the density of Co in the computational model. The factor of reaction temperature should also be carefully considered to assess how much the reaction rate of rate limiting step can be altered by a decrease of activation energy by 0.04 eV (Fig 5c).

4. The authors reported a monotonic decrease of the spin moment of Co from 0.6 to 0 as the number of Co sites increases (Fig 3d), which is surprising for me. As more Co atoms anchored in the N/O doped graphene → lower valence state of Co (indicated by less Bader charge), how the spin moment of Co ion changes depends on the specific value of valence number and the strength of coordination field, but it is hard to imagine that the spin moment of Co almost disappears. The authors cited ref 20 to support their explanation of change of spin moment, which also puzzles me. Because in Ref 20, FeN₃/G always carries a non-zero spin moment no matter the entire system is ferromagnetic or antiferromagnetic.

5. The computation procedure for determining the magnetic ground state of different Co SACs should be stated clearly in the computational detail section.

6. The specific value of U in DFT+U calculations with the reference paper should be provided explicitly.

7. It is also surprising for me that the trans-stilbene (SB) adsorbed on the catalyst would not interact with two adjacent Co sites simultaneously, considering SB is large enough to cover two Co sites (Fig 5c).

8. The value of imaginary frequency of the located transition state by CINEB calculation should be reported.

9. Phytic acid was used in the synthesis of Co SACs. Why there is no P in the obtained catalyst?

10. The reason why densely populated Co SACs weaken the adsorption of the SBO molecule should be explained.

11. The resolution of Figure 5c should be increased.

12. The y-axis and caption for Fig 3c used "d-bond center", which should be corrected.

Reviewer #2 (Remarks to the Author):

In this work, the authors report a detailed study about densely populated Co single-atom catalysts for the enhanced alkene epoxidation. Developing novel catalytic systems based on 3d-metal heterogeneous

materials that are highly efficient and selective for organic reactions remains a crucial research area in both academic laboratories and industries. This manuscript also presents detailed structural characterization, insights into the role of increased metal density in regulating the electronic states of single metal centers, and mechanistic understanding of the enhanced catalytic performance. This work is highly interesting and I would recommend it for publication after addressing the following issues.

(1) The authors have demonstrated that the densely populated Co single-atom catalysts exhibit significantly better catalytic performance compared to Co NPs. However, further characterization and interpretation are required to explain the failure of Co NPs in trans-stilbene epoxidation

(2) Identifying the Co-N3O1 coordination structure can be challenging, as it is often difficult to differentiate between nitrogen and oxygen coordination. The authors need to explain how to identify the Co-N3O1 coordination structure.

(3) While the authors focused on studying Co single-atom catalysts for alkene epoxidation, it would be interesting to investigate the performance of other transition metal single-atom catalysts and how they compare?

(4) Is there a change in the morphology and local structure of the support with increasing metal content?

(5) Comparing the Turnover Frequency (TOF) and Turnover Number (TON) values could further demonstrate the advantages of densely populated single-atom catalysts.

(6) It is suggested to expand the substrate scope of the alkene epoxidation reaction.

(7) The synthesis details of Co1/NOC-11 and Co NPs are missing.

(8) The NMR data shall be presented in a standard form (National Science Review, 2022, nwac100, doi.org/10.1093/nsr/nwac100, Nature communications 9 (1), 3197, 2018), please present raw data rather than plotted lines.

Reviewer #3 (Remarks to the Author):

Cao and co-authors report a new finding about the effect of single-atom density on regulating the electronic structures of metal SACs and their catalytic performances for alkene epoxidation. They also clarify the mechanism of the enhanced catalytic performance from both experimental and theoretical calculations by taken Co as an example. That is, the electronic structure of Co atoms in densely populated Co SACs is altered through charge redistribution, resulting in less Bader charger and higher d-band center, which are beneficial for activation of O₂ and trans-stilbene as well as product desorption.

In my opinion, this manuscript is very interesting and the following points should be elaborated:

- (1) Is the N3O1 vacancy in Figure 3a real existence in experiments? These vacancy-models generally result in electron deficiency/rich that possibly affect the O₂ adsorption. As a result, the O₂ adsorption on defective N or near C atom site should be examined.
- (2) The calculations of 2-Co1-N3O1 in Figure 5 were missing.
- (3) The ferromagnetic state of Co site is closely related to the O₂ adsorption (J. Am. Chem. Soc. 143 (2021) 4405-4413) and the corresponding the ground state could be changed. Therefore, the effect of O₂ adsorption on ferromagnetic state should be investigated.
- (4) As to the reaction mechanism, the initial point is possibly the *O₂ or *SB. The two benzene rings in SB are likely to interact with the catalyst by π - π interaction. The Co1+*O₂+*SB showed in the current manuscript maybe not in the stable state.
- (5) The van der Waals interactions between catalyst and reactants should be considered in DFT calculations, especially for the adsorption energies of structures of Co1+*O₂+*SB, Co1+*O+*SB, Co1+*SBO, and the correspondingly related activity barriers.
- (6) How to make sure the same coordination structures in all Co SACs with varying densities? Did the support maintain the same with increasing metal loadings? The authors should give more discussion about it.
- (7) How about the catalytic performances of other transition metal SACs for alkene epoxidation? Similarly, other substrates should be test to further demonstrate the usefulness of densely populated Co SAC.
- (8) In the 2-Co1-N3O1 and 4-Co1-N3O1 models, how about the Bader charge of every C single atom? The Bader charge values should be provided.
- (9) What's the difference about Co NPs for alkene epoxidation?
- (10) Fig. 1f is not visible, please verify it.

Response to reviewers

Reviewer #1 (Remarks to the Author):

Jin et al. reported a combined experimental and computational study of Co single atom catalysts (SACs) showing that the electronic structures in terms of charge distribution can be regulated by the active site density and increasing that density can lead to enhanced catalytic performance in alkene epoxidation. In the computation part, they found less Bader charge and higher d-band center for Co SACs as the density of Co increases. As a result, higher density Co SACs can bind O₂ stronger and activate it to a larger extent. The desorption of the product SBO is also easier. Then the authors explain the experimentally observed enhancement of alkene epoxidation using the computed electronic structure of catalysts and free energy profiles of reactions. Overall, this is a comprehensive and interesting study of the impact of density of SACs on a specific catalytic reaction. However, the significance of this work and the novelty of the mechanistic understanding are not that strong to meet the requirement of publication in Nature Communications. Therefore, I could not recommend its publication in the current form. Specific comments that need to be addressed are as follows:

Answer: Thank you very much for your comments and valuable suggestions, which are helpful for us to improve the manuscript quality. We revised the manuscript carefully according to your advice.

1. The statement in the abstract as “such an interaction has rarely been studied and its influence on the reaction mechanisms in various reactions remains unclear” (Line 19-20) and another one in the introduction as “(for densely populated SACs) further influence the local geometric or electronic structure of individual metal centers via electron transfer, spin coupling or charge redistribution, and thus affect the intrinsic activity of active sites” (Line 50-52) are not on the same page. As a matter of fact, inter-site interactions and their influence on mechanisms of selected reactions have been studied in a couple of previous work of SACs, including theoretical (ref 20) and experimental studies (ref 21) that the authors have cited, and a few more, like Nano. Lett. 2022, 22, 3744; JACS Au 2021, 1, 2130 (review paper); JPCL 2021, 12, 5233; JPCL 2020, 11, 3962; ACS AMI 2020, 12, 15271; JPCL 2019, 10, 7009. It is true that there is no unified theoretical framework to predict how they influence various reactions, but this work also just focused on one specific type of reaction -- alkene epoxidation. The authors should elaborate the background of their work more accurately.

Answer: Thank you very much for your comments and suggestion. Indeed, the inter-site interactions and their influence on catalytic performance in various reactions have been studied in a couple of previous works of SACs. However, the underlying mechanisms were distinct, and there was no unified theoretical guidance available at the time. Therefore, much more effort is needed to investigate such an interaction in various reactions, which will give a deeper mechanistic understanding of structure-performance relationship at the atomic scale.

The main novelty of this work lies in the following two aspects: (1) we report a general and reliable strategy to synthesize various densely populated M-SACs (M=Fe, Co, Ni, Cu, Zn, Ru, and Ir) with loading up to 35.5 wt%, and investigate the effect of metal density on the single-atom electronic structure systematically. (2) We present a new finding about the site interaction in Co SACs for alkene epoxidation. To our knowledge, there is the first report

about inter-site interaction of SACs for alkene epoxidation reaction. Moreover, we reveal the underlying mechanism of how the site interaction influences the electronic structure of Co atoms and catalytic reaction, which also is not disclosed before.

We revised the background of our work in the revised manuscript.

2. Following the first comment, I am afraid that the novelty of this work is not clearly stated, either in the experimental side, like alternative synthetic strategy of making very high metal loading SACs or much higher yield of alkene epoxidation than other work, or in computational side, like new mechanism that has never been reported before. It is noted that this work is similar to their just published work (ref 28. ACS Catalysis 2023, 13, 1316–1325) of which the title is “Understanding the Density-Dependent Activity of Cu Single-Atom Catalyst in the Benzene Hydroxylation Reaction”. What is the advance provided exclusively in this work?

Answer: Thank you very much for your comment and suggestion. In our just published work, we aimed to solve two key problems in the benzene hydroxylation reaction: poor H₂O₂ utilization efficiency (usually < 10%) and low mass loading (generally < 3 wt %). We found the maximum H₂O₂ utilization efficiency can surpass 50% by using ultra-high density Cu SAC of 2.4 atoms/nm², which offers a new view of preparing efficient SACs for phenol production.

The main novelty of this work lies in the following two aspects: (1) we report a general and reliable strategy to synthesize various densely populated M-SACs (M=Fe, Co, Ni, Cu, Zn, Ru, and Ir) with loading up to 35.5 wt%, and investigate the effect of metal density on the single-atom electronic structure systematically. (2) We present a new finding about the site interaction in Co SACs for alkene epoxidation. To our knowledge, there is the first report about inter-site interaction of SACs for alkene epoxidation reaction. Moreover, we reveal the underlying mechanism of how the site interaction influences the electronic structure of Co atoms and catalytic reaction, which also is not disclosed before.

3. Regarding to the mechanistic understanding, I think the advance reported in this work is probably not impressive as measured by the standard of Nature Communications. My major concern is that the connection between the tuning of electronic structures due to the increase of active site density and the appreciable improvement of experimental yield owing to higher metal loading in SACs is not strong. I did not see clear evidence to support a hypothesis that the former (in a microscopic level) causes the latter (in a macroscopic level). Since they have calculated the whole free energy profile including activation barrier for all elementary reactions, they may calculate TOF to see how much the rate can be altered due to the increase of the density of Co in the computational model. The factor of reaction temperature should also be carefully considered to assess how much the reaction rate of rate limiting step can be altered by a decrease of activation energy by 0.04 eV (Fig 5c).

Answer: Thank you very much for your valuable suggestion. Firstly, we carried out DFT calculations with addition of van der Waals correction in view of the possible intense π - π interaction between SB and catalyst. As shown in Fig. 5c, the transition barriers of rate determining step (RDS) in 1-Co, 2-Co, and 4-Co were determined to be 0.79, 0.71 and 0.61 eV, respectively (Fig. 5c), which became larger after correction.

Secondly, according to your suggestion, we evaluated the effect of reaction temperature on the catalytic performance of Co₁/NOC-x (Supplementary Fig. 24). According to the Arrhenius

equation, the apparent activation energy of Co₁/NOC-21 was calculated to be about 58.2 kJ·mol⁻¹, which was much lower than those of Co₁/NOC-11 (67.7 kJ·mol⁻¹) and Co₁/NOC-5 (71.2 kJ·mol⁻¹), confirming lower energy barrier and much higher catalytic activity of high-density Co₁/NOC-21.

Thirdly, we calculated TOF values of Co₁/NOC-x samples in alkene epoxidation. As shown in Fig. 4c, Co₁/NOC-21 exhibited the highest TOF value of 96.7 h⁻¹, which was 10 times than that of Co₁/NOC-5.

All above experimental and DFT calculation results agreed well and suggested the intrinsic high activity of densely populated Co₁/NOC-21 catalyst. We added these discussions to the revised manuscript.

Fig. 5 (c) Energy profiles of trans-stilbene epoxidation reaction over 1-Co₁-N₃O₁, 2-Co₁-N₃O₁ and 4-Co₁-N₃O₁ models. Inset: the configurations of intermediates of 4-Co₁-N₃O₁.

Fig. 4 (c) TOF and mass-specific activity (MSA) of various Co₁/NOC samples.

Supplementary Fig. 24 SB conversion against reaction time with various samples of Co₁/NOC-21 (a), Co₁/NOC-11 (b), and Co₁/NOC-5 (c) under different reaction temperatures. (d) Arrhenius plots and corresponding apparent activation energies of the Co₁/NOC-x catalysts.

4. The authors reported a monotonic decrease of the spin moment of Co from 0.6 to 0 as the number of Co sites increases (Fig 3d), which is surprising for me. As more Co atoms anchored in the N/O doped graphene → lower valence state of Co (indicated by less Bader charge), how the spin moment of Co ion changes depends on the specific value of valence number and the strength of coordination field, but it is hard to imagine that the spin moment of Co almost disappears. The authors cited ref 20 to support their explanation of change of spin moment, which also puzzles me. Because in Ref 20, FeN₃/G always carries a non-zero spin moment no matter the entire system is ferromagnetic or antiferromagnetic.

Answer: Thank you for your comment and suggestion. We re-did DFT+U calculations with setting the U value of 1.0 according to several reference papers (J. Magn. Magn. Mater. 2022, 549, 169005; Phys. Rev. Lett. 2014, 113, 175502; Phys. Rev. B 2021, 103, 094445; J. Magn. Magn. Mater. 2022, 556, 169396; Mater. Sci.-Pol. 2017, 35(4), 846-856; Phys. Rev. B, 2012, 86, 2, 024435). Besides, we carried out the calculation on the spin moments of Co in 4-Co-N3O1 by hybrid density functional (HSE06) as well, and check the reasonability of U value. The average spin moments of Co obtained from DFT+U calculations using U = 1 was 0.310 μ_B, very closed to that given by HSE06 (0.254 μ_B). It demonstrated that valuing U = 1 could give more reasonable results of spin moments.

As shown in Supplementary Table 7, the average spin moment of Co drops from 0.797 to 0.310 as the number of Co sites increases. This result suggests that there is no decisive

relationship with the specific value of valence number, considering the Co atoms in different catalysts have similar valence states indicated by Bader charge (the difference is 0.03 e between 1-Co₁-N₃O₁ and 4-Co₁-N₃O₁). According to the pDOS profiles in Fig. 3b and Supplementary Fig. 23, the Co 3d pDOS displayed a certain degree of asymmetric distribution between up-spin and down-spin pDOS. Meanwhile, the spin level splitting to some extent was also observed, as more part of down-spin pDOS was situated above Fermi level. The more aggregation in up-spin pDOS induced by above two reasons resulted in the final spin moments. The decrease of spin moment from 1-Co₁-N₃O₁ to 4-Co₁-N₃O₁ can be ascribed to the down shift of energy of the Co 3d orbital as adjacent Co atoms get closer (Fig. 3d), which was consistent with the reported result (Nat. Catal. 2021, 4, 615-622).

We added these discussions to the revised manuscript.

Supplementary Table 7. The spin moments of Co atoms in different x-Co₁-N₃O₁ models.

Subs	Co label	1-Co ₁ -N ₃ O ₁	2-Co ₁ -N ₃ O ₁	4-Co ₁ -N ₃ O ₁
spin moment (μ_B)	Co1	0.797	0.438	0.319
	Co2		-0.412	-0.306
	Co3			-0.311
	Co4			0.304
Average (μ_B)	Co	0.797	0.425	0.310

Supplementary Fig. 23 The corresponding spin moment originated from the spin-splitting of Co 3d atom orbitals in x-Co₁-N₃O₁ models.

Fig. 3 (d) The plot of spin moment with respect to Co single-atom density. Inset: The spin density isosurfaces of $0.005 \text{ e}/\text{\AA}^3$.

5. The computation procedure for determining the magnetic ground state of different Co SACs should be stated clearly in the computational detail section.

Answer: Thank you for your kind suggestion. We added the computation procedure for determining the magnetic ground state in the revised computational detail section.

6. The specific value of U in DFT+U calculations with the reference paper should be provided explicitly.

Answer: Thank you for your suggestion. The U value for Co used in DFT+U calculations in this work was set as 1.0 according to several reference papers (J. Magn. Magn. Mater. 2022, 549, 169005; Phys. Rev. Lett. 2014, 113, 175502; Phys. Rev. B 2021, 103, 094445; J. Magn. Magn. Mater. 2022, 556, 169396; Mater. Sci.-Pol. 2017, 35(4), 846-856; Phys. Rev. B, 2012, 86, 2, 024435).

Besides, we carried out the calculation on the spin moments of Co in 4-Co-N₃O₁ by hybrid density functional (HSE06) as well, and check the reasonability of U value. The average spin moments of Co obtained from DFT+U calculations using U = 1 was 0.310 μ_B , very closed to that given by HSE06 (0.254 μ_B). It demonstrated that valuing U = 1 could give more reasonable results of spin moments.

7. It is also surprising for me that the trans-stilbene (SB) adsorbed on the catalyst would not interact with two adjacent Co sites simultaneously, considering SB is large enough to cover two Co sites (Fig 5c).

Answer: Thank you for your comment and valuable question. According to your suggestion, we calculated the energy changes of the reaction in which the two benzene rings interact with two adjacent Co sites simultaneously (Supplementary Fig. 37). The energy barrier for the oxidation of benzene ring by two adjacent Co sites was calculated to be 2.02 eV (formation of epoxy benzene) or 2.60 eV (formation of phenol), which was much higher than the corresponding value of the SB exoxidation path (0.61 eV, Fig. 5c). Therefore, the interaction

of SB with two adjacent Co sites can be excluded as the main reaction path in SB epoxidation. We added above results and discussions to the revised manuscript.

Supplementary Fig. 37 The reaction pathways for formation of (a) epoxy benzene and (b) phenol. Inset: the corresponding configurations of intermediates on 4-Co₁-N₃O₁.

8. The value of imaginary frequency of the located transition state by CINEB calculation should be reported.

Answer: Thank you for your suggestion. As shown in Supplementary Table 10, the sole value of imaginary frequencies of TS-I and TS-II on 1-Co₁-N₃O₁, 2-Co₁-N₃O₁ and 4-Co₁-N₃O₁ are 427.38 and 567.23 cm⁻¹, 613.61 and 626.22 cm⁻¹, 509.13 and 400.47 cm⁻¹, respectively. We added it to the revised ESI.

Supplementary Table 10. The sole value of imaginary frequencies of TS-1 and TS-2 on various x-Co₁-N₃O₁ models.

Subs	1-Co ₁ -N ₃ O ₁	2-Co ₁ -N ₃ O ₁	4-Co ₁ -N ₃ O ₁
TS-I (cm ⁻¹)	427.38	613.61	509.13
TS-II (cm ⁻¹)	567.23	626.22	400.47

9. Phytic acid was used in the synthesis of Co SACs. Why there is no P in the obtained catalyst?

Answer: Thank you for your question. We investigated the coordination structure of as-prepared Co SACs through EXAFS. It found that there was no Co-P first-coordination shell (Supplementary Fig. 19). These P atoms might be located at second or higher coordination shell. However, it is really difficult to identify the precise location with current characterization tools. It is widely known that the activity of SACs is mainly dominated by the first-coordination shell. Thus, we didn't consider P atoms in the DFT calculations.

In addition, in order to exclude the activity contribution of P atoms, we evaluated the catalytic performance of P-doped carbon support alone in trans-stilbene epoxidation. It found that the P-doped support showed negligible activity (Fig. 4b). Therefore, we can reasonably exclude the influence of P atoms on the catalytic activity.

We added these discussions to the revised manuscript.

Supplementary Fig. 19 k^3 -weight FT-EXAFS spectra of $\text{Co}_1/\text{NOC-21}$. Curves from top to bottom are the Co-P, Co-N backscattering pathways.

Fig. 4 (b) The dynamics plots of trans-stilbene conversion against reaction time over various samples. Reaction condition: 1 mmol trans-stilbene epoxidation, containing 0.01 mmol Co in each catalyst (or 10 mg support), 10 mL solvent, 140 °C, O_2 balloon.

10. The reason why densely populated Co SACs weaken the adsorption of the SBO molecule should

be explained.

Answer: Thank you very much for your question. We carried out DFT calculations with addition of van der Waals correction in view of the possible intense π - π interaction between SB and catalyst. After correction, the adsorption of SBO molecule on the three substrates showed similar intensities of 0.31, 0.33 and 0.34 eV, respectively.

We corrected the statement in the revised manuscript.

11. The resolution of Figure 5c should be increased.

Answer: We have updated Fig. 5c with high resolution in revised manuscript.

12. The y-axis and caption for Fig 3c used “d-bond center”, which should be corrected.

Answer: Thank you for your carefully reading. We have corrected it in the revised manuscript.

Reviewer #2 (Remarks to the Author):

In this work, the authors report a detailed study about densely populated Co single-atom catalysts for the enhanced alkene epoxidation. Developing novel catalytic systems based on 3d-metal heterogeneous materials that are highly efficient and selective for organic reactions remains a crucial research area in both academic laboratories and industries. This manuscript also presents detailed structural characterization, insights into the role of increased metal density in regulating the electronic states of single metal centers, and mechanistic understanding of the enhanced catalytic performance. This work is highly interesting and I would recommend it for publication after addressing the following issues.

Answer: Thank you very much for your positive comments and valuable suggestions, which are helpful for us to improve the manuscript quality. We revised the manuscript carefully according to your advice.

1. The authors have demonstrated that the densely populated Co single-atom catalysts exhibit significantly better catalytic performance compared to Co NPs. However, further characterization and interpretation are required to explain the failure of Co NPs in trans-stilbene epoxidation

Answer: Thank you for your suggestion. We used DFT calculations to understand the trans-stilbene epoxidation reaction pathways of Co NPs. Both XRD and HRTEM characterizations revealed that Co NPs exposed with Co(111) crystal faces (Supplementary Fig. 38-39). Therefore, we used Co(111) to represent Co NPs/NC sample for calculations. As shown in Supplementary Fig. 40, an O₂ molecule was adsorbed and dissociated to O* on Co(111) with a high exothermic of - 4.56 eV (III). The SB molecule was then activated and converted to SBO with an energy barrier of 1.21 eV (IV-V). By overcoming another energy barrier of 1.36 eV (V to VI), the remaining oxygen can be transformed to a new one-coordinated oxygen atom and subsequently participate in the next step of the SB epoxidation cycle. The highest energy barrier for SBO production reached to 1.78 eV (VII to VIII). It can be seen that the energy barrier on Co NPs was much higher than on Co1/NOC-x samples, resulting in the lowest reaction rate in trans-stilbene epoxidation, which was consistent with experimental results.

We added these discussions to the revised manuscript.

Supplementary Fig. 38 XRD pattern of Co NPs/NC sample.

Supplementary Fig. 39 (a) TEM and (b) HRTEM images of Co NPs/NC sample. (c) HAADF image and corresponding EDS distribution mapping.

Supplementary Fig. 40 Energy profiles of trans-stilbene epoxidation reaction on Co(111) of Co NPs/NC models. Inset: the configurations of intermediates.

2. Identifying the Co-N₃O₁ coordination structure can be challenging, as it is often difficult to differentiate between nitrogen and oxygen coordination. The authors need to explain how to identify the Co-N₃O₁ coordination structure.

Answer: Indeed, C/N/O cannot be well-distinguished by EXAFS. Most reported studies generally used the EXAFS simulation to reveal the N/O dual-coordination structures (Adv. Mater. 2021, 33, 2107103; Angew. Chem. Int. Ed. 2022, 61, e202202338; Small 2023, 19, 2205583). We also first employed the EXAFS simulation to identify the $\text{Co}_1\text{-N}_3\text{O}_1$ coordination structure. As shown in Fig. 2g, the simulation curves matched very well with the experimental curves, indicate the highly reliability of the $\text{Cu}_1\text{-N}_3\text{O}_1$ configuration. Moreover, element analysis showed that atom ratios of N to O were kept as ~ 3 in all $\text{Co}_1/\text{NOC-x}$ (Supplementary Table 5).

We added these discussions to the revised manuscript.

3. While the authors focused on studying Co single-atom catalysts for alkene epoxidation, it would be interesting to investigate the performance of other transition metal single-atom catalysts and how they compare?

Answer: Thank you for your suggestion. We investigated the performance of other transition metal single-atom catalysts for alkene epoxidation. Co_1/NOC SAC can convert up to 96.5% of SB after 1 h, but other transition metal SACs can only convert up to 54.4%, showing that Co SACs have the highest catalytic performance in SB epoxidation (Fig. 4a).

We added this result to the revised manuscript.

Fig. 4 (a) The catalytic performance of various transition metal M_1/NOC samples in trans-stilbene epoxidation within 1 h.

4. Is there a change in the morphology and local structure of the support with increasing metal content?

Answer: We performed SEM, Raman and XPS to investigate the morphology and structure of the supports. As shown in Supplementary Fig. 13, typical SEM images show that all supports maintain nanosheet morphology. The Raman and C 1s XPS spectra of $\text{Co}_1/\text{NOC-x}$ demonstrate a similar local structure of supports (Supplementary Fig. 20).

We added above results to the revised manuscript.

Supplementary Fig. 13 SEM images of 5.4 wt%, 10.9 wt%, and 21.2 wt% Co SACs samples.

Supplementary Fig. 20 (a) Raman spectra and (b) C 1s XPS spectra of Co₁/NOC-x samples.

5. Comparing the Turnover Frequency (TOF) and Turnover Number (TON) values could further demonstrate the advantages of densely populated single-atom catalysts.

Answer: Thank you for your suggestion. We calculated TOF values of various samples in alkene epoxidation. As shown in Fig. 4c, Co₁/NOC-21 exhibited the highest TOF value of 96.7 h⁻¹, which was 10 times than that of Co₁/NOC-5. Particularly, the mass-specific activity is more important for industrial applications. Due to the high metal loading and intrinsic activity, the mass-specific activity of Co₁/NOC-21 reached as high as 193 mol·g⁻¹·h⁻¹, which was nearly 30 times and even 15 times to that of Co₁/NOC-5 and Co NPs, respectively. Such high mass-specific activity was also much higher than the values reported in literature, demonstrating the advantages of densely populated SACs.

We added above discussions to the revised manuscript.

Fig. 4 (c) TOF and mass-specific activity (MSA) of various Co₁/NOC samples.

6. It is suggested to expand the substrate scope of the alkene epoxidation reaction.

Answer: Thank you very much for your suggestion. We examined a broad scope of substrates in the alkene epoxidation reaction, as shown in Supplementary Table 8. These epoxides are valuable intermediates for the preparation of organic molecules in industry. It was found that the substituted trans-stilbene derivatives (Entries 1-4) and other aromatic tri-, and mono-substituted alkenes (Entries 5-7) can be epoxidized to the desired products with high efficiency.

**This suggests Co₁/NOC-21 can tolerate a broad scope of substrates in alkene epoxidation.
We added these results to the revised manuscript.**

Supplementary Table 8. Substrate scope of alkene epoxidation over the Co₁/NOC-21 catalyst.

Entry	Substrate	Production	Time (h)	Conv. (%)	Sel. (%)
1			1	96.5	98.1
2			1.5	97.8	99.2
3			5	91.3	98.5
4			3	99	97.3
5			7	>99	>99
6			1	76.1	78.9
7			2.5	84.9	97.2

7. The synthesis details of Co₁/NOC-11 and Co NPs are missing.

Answer: Thank you for your carefully reading. We added the synthesis details of Co₁/NOC-11 and Co NPs in the revised ESI.

8. The NMR data shall be presented in a standard form (National Science Review, 2022, nwac100, doi.org/10.1093/nsr/nwac100, Nature communications 9 (1), 3197, 2018), please present raw data rather than plotted lines.

Answer: Thank you for your suggestion. We presented the NMR data in a standard form in the revised manuscript.

Reviewer #3 (Remarks to the Author):

Cao and co-authors report a new finding about the effect of single-atom density on regulating the electronic structures of metal SACs and their catalytic performances for alkene epoxidation. They also clarify the mechanism of the enhanced catalytic performance from both experimental and theoretical calculations by taken Co as an example. That is, the electronic structure of Co atoms in

densely populated Co SACs is altered through charge redistribution, resulting in less Bader charge and higher d-band center, which are beneficial for activation of O₂ and trans-stilbene as well as product desorption. In my opinion, this manuscript is very interesting and the following points should be elaborated:

Answer: Thank you very much for your positive comments and valuable suggestions, which are helpful for us to improve the manuscript quality. We revised the manuscript carefully according to your advice.

1. Is the N₃O₁ vacancy in Figure 3a real existence in experiments? These vacancy-models generally result in electron deficiency/rich that possibly affect the O₂ adsorption. As a result, the O₂ adsorption on defective N or near C atom site should be examined.

Answer: Indeed, C/N/O cannot be well-distinguished by EXAFS. Most reported studies generally used the EXAFS simulation to reveal the N/O dual-coordination structures (Adv. Mater. 2021, 33, 2107103; Angew. Chem. Int. Ed. 2022, 61, e202202338; Small 2023, 19, 2205583). We also employed the EXAFS simulation to identify the Co₁-N₃O₁ coordination structure. As shown in Fig. 2g, the simulation curves matched very well with the experimental curves, indicate the highly reliability of the Cu₁-N₃O₁ configuration. Moreover, element analysis showed that atom ratios of N to O were kept as ~ 3 in all Co₁/NOC-x (Supplementary Table 5). These results indicate that the existence of Co₁-N₃O₁ coordination structure.

According to your suggestion, we then examined the possibility of O₂ adsorption on defective N or near C atom site. The calculated O-O bond distance after adsorption on vacancy was only 1.25 Å (equal to that of free O₂), implying O₂ molecules cannot be activated on defective N₃O₁ vacancies (Supplementary Fig. 31). In contrast, O-O bond was enlarged to 1.29 Å after adsorption on the Co atom in 1-Co₁-N₃O₁ model. These results indicate that O₂ molecules should be adsorbed and activated on Co atoms during the SB expoxidation.

We added these discussions to the revised manuscript.

Supplementary Fig. 31 The O-O bond after adsorption on Co atom and vacancy sites.

2. The calculations of 2-Co₁-N₃O₁ in Figure 5 were missing.

Answer: Thank you for your reminding. According to your suggestion, we did DFT calculations of 2-Co₁-N₃O₁ for further comparison. As shown in Fig. 5, all the results show that it agreed well with the trend among Co₁-N₃O₁ models, confirming the rationality of the proposed reaction mechanism.

We added these results and discussions to the revised manuscript.

Fig. 5 Theoretical calculation of adsorption configurations and reaction pathways. (a) The calculated energies for adsorption O₂ and SB, as well as SBO desorption energies. (b) Charge density differences after O₂ adsorption on various x-Co₁-N₃O₁ models. (c) Energy profiles of trans-stilbene epoxidation reaction over 1-Co₁-N₃O₁, 2-Co₁-N₃O₁ and 4-Co₁-N₃O₁ models. Inset: the configurations of intermediates of 4-Co₁-N₃O₁.

3. The ferromagnetic state of Co site is closely related to the O₂ adsorption (*J. Am. Chem. Soc.* 143 (2021) 4405-4413) and the corresponding the ground state could be changed. Therefore, the effect of O₂ adsorption on ferromagnetic state should be investigated.

Answer: Thank you very much for this important advice. According to your suggestion, the ferromagnetic states of Co atoms and the spin states after O₂ adsorption were investigated. As shown in Supplementary Fig. 33 and Supplementary Table 9, it was found that the spin moment of the neighboring Co atom in 2-Co₁-N₃O₁ was even reversed after O₂ adsorption, suggesting the ground state was changed (*J. Am. Chem. Soc.* 2018, 140, 45, 15149–15152). In addition, there was more pDOS overlap and a larger energy splitting between bonding and antibonding orbitals (Supplementary Fig. 34). 4-Co₁-N₃O₁ showed the largest spin moment of 0.443 μB. Meanwhile, the adsorbed O₂ also exhibited the largest atomic or molecular spin moments, which was consistent with previous work (*J. Am. Chem. Soc.* 2021, 143, 4405-4413). The larger variations in spin moments of Co and O₂ in 4-Co₁-N₃O₁ before and after oxygen adsorption demonstrate the most intensive electron transfer from Co to O₂ (Fig. 5b). As a consequence, when O₂ was adsorbed on high-density Co with the lowest activation energy, the biggest increase in the O-O bond distance occurred.

We added above discussions to the revised manuscript.

Supplementary Fig. 33 The spin density isosurfaces of various $x\text{-Co}_1\text{-N}_3\text{O}_1$ models. Isosurfaces: $0.003 \text{ e}/\text{\AA}^3$.

Supplementary Fig. 34 The corresponding spin moment originated from the spin-splitting of Co 3d and O 2p orbitals after O_2 adsorption on various $x\text{-Co}_1\text{-N}_3\text{O}_1$ models.

Supplementary Table 9. The spin moments of O and Co atoms in different O_2 -adsorbed $x\text{-Co}_1\text{-N}_3\text{O}_1$ models.

Subs	Label	1- $\text{Co}_1\text{-N}_3\text{O}_1$	2- $\text{Co}_1\text{-N}_3\text{O}_1$	4- $\text{Co}_1\text{-N}_3\text{O}_1$
spin moment (μ_B)	Co1	0.088	0.159	0.443
	Co2		-0.269	0.149
	Co3			-0.114
	Co4			0.163
	O1	0.362	0.387	0.406
	O2	0.430	0.434	0.435
	Total in O_2	0.792	0.824	0.841

4. As to the reaction mechanism, the initial point is possibly the $^*\text{O}_2$ or $^*\text{SB}$. The two benzene rings in SB are likely to interact with the catalyst by π - π interaction. The $\text{Co}_1+^*\text{O}_2+^*\text{SB}$ showed in the current manuscript maybe not in the stable state.

Answer: As we replied in question 1, O_2 adsorption on the N_3O_1 vacancy and Co site was proved to be thermodynamic preferential. So all the vacancies and Co sites was covered by O_2 molecule through the whole reaction. The $\text{Co}_1+^*\text{O}_2+^*\text{SB}$ was calculated to be in a stable state with the free energy of $-1.39 \text{ eV} \sim -1.58 \text{ eV}$ in three models (Supplementary Table 11).

5. The van der Waals interactions between catalyst and reactants should be considered in DFT calculations, especially for the adsorption energies of structures of $\text{Co}_1+\text{*O}_2+\text{*SB}$, $\text{Co}_1+\text{*O}+\text{*SB}$, $\text{Co}_1+\text{*SBO}$, and the correspondingly related activity barriers.

Answer: Thank you very much for your valuable advice. According to your suggestion, the van der Waals correction was adopted in our modified calculations. And it was revealed that the van der Waals interaction indeed affected the energies about the adsorption/desorption of SB/SBO (Supplementary Table 11). In detail, all the adsorption processes had more negative values, while the desorption had more positive values, possibly attributed to the π - π interaction between catalyst and SB/SBO. However, it didn't change the rate determining step (Fig. 5c), as the highest barriers (TSI) were still located between $\text{Co}_1+\text{*O}_2+\text{*SB}$ and $\text{Co}_1+\text{*O}+\text{*SBO}$. The correspondingly related activity barriers of rate determining step in 1, 2, 4- $\text{Co}_1\text{-N}_3\text{O}_1$ models were calculated to be 0.79, 0.71 and 0.61 eV, respectively.

We revised the results in the revised manuscript and ESI.

Supplementary Table 11. Free energies of the reaction intermediates with different $\text{Co}_1\text{-N}_3\text{O}_1$ models.

Step	Model (eV)		
	1- $\text{Co}_1\text{-N}_3\text{O}_1$	2- $\text{Co}_1\text{-N}_3\text{O}_1$	4- $\text{Co}_1\text{-N}_3\text{O}_1$
Co_1	0	0	0
Co_1+O_2^*	-1.10	-1.12	-1.15
$\text{Co}_1+\text{O}_2^*+\text{SB}^*$	-1.55	-1.48	-1.39
TS-I	-0.76	-0.77	-0.78
$\text{Co}_1+\text{O}^*+\text{SBO}^*$	-1.93	-1.98	-1.97
Co_1+O^*	-1.50	-1.47	-1.49
$\text{Co}_1+\text{O}^*+\text{SB}^*$	-2.07	-2.02	-2.04
TS-II	-1.70	-1.67	-1.86
Co_1+SBO^*	-2.93	-2.98	-2.98
Co_1+SBO	-2.62	-2.65	-2.64

6. How to make sure the same coordination structures in all Co SACs with varying densities? Did the support maintain the same with increasing metal loadings? The authors should give more discussion about it.

Answer: Thank you for your suggestion. A two-step strategy incorporating polycondensation and subsequent pyrolysis procedures was developed to synthesize the densely populated metal SACs. The key success of this strategy relies on the controllable polycondensation during the first step, whereby the small molecules (melamine, cyanuric acid, L-alanine and phytic acid) are spontaneously polymerized in water to form two-dimensional nanosheets (Supplementary

Fig. 2). At the same time, metal ions are complexed in it by the surrounding heteroatoms (N/O). Metal SACs with N/O-coordination structure are produced after pyrolysis at moderate temperature (700 °C) in an Ar atmosphere. By only adjusting the amount of Co precursor, a series of Co SACs with varying densities can be obtained.

In addition, we performed SEM, Raman and XPS to investigate the morphology and structure of the support. As shown in Supplementary Fig. 13, typical SEM images show that all supports maintain nanosheet morphology. The Raman and C 1s XPS spectra of Co₁/NOC-x demonstrate a similar local structure of supports (Supplementary Fig. 20). These results suggest the support should maintain the same with increasing metal loadings.

We added above results to the revised manuscript.

Supplementary Fig. 20 (a) Raman spectra and (b) C 1s XPS spectra of Co₁/NOC-x samples.

7. How about the catalytic performances of other transition metal SACs for alkene epoxidation? Similarly, other substrates should be test to further demonstrate the usefulness of densely populated Co SAC.

Answer: Thank you for your suggestion. We investigated the performance of other transition metal single-atom catalysts for alkene epoxidation. Co₁/NOC SAC can convert up to 96.5% of SB after 1 h, but other transition metal SACs can only convert up to 54.4%, showing that Co SACs have the highest catalytic performance in SB epoxidation (Fig. 4a).

We then examined a broad scope of substrates in the alkene epoxidation reaction, as shown in Supplementary Table 8. These epoxides are valuable intermediates for the preparation of organic molecules in industry. It was found that the substituted trans-stilbene derivatives (Entries 1-4) and other aromatic tri-, and mono-substituted alkenes (Entries 5-7) can be epoxidized to the desired products with high efficiency. This suggests Co₁/NOC-21 can tolerate a broad scope of substrates in alkene epoxidation.

We added these results to the revised manuscript.

Fig. 4 (a) The catalytic performance of various transition metal M_1 /NOC samples in trans-stilbene epoxidation within 1 h.

Supplementary Table 8. Substrate scope of alkene epoxidation over the Co1/NOC-21 catalyst.

Entry	Substrate	Production	Time (h)	Conv. (%)	Sel. (%)
1			1	96.5	98.1
2			1.5	97.8	99.2
3			5	91.3	98.5
4			3	99	97.3
5			7	>99	>99
6			1	76.1	78.9
7			2.5	84.9	97.2

8. In the 2-Co1-N3O1 and 4-Co1-N3O1 models, how about the Bader charge of every Co single atom? The Bader charge values should be provided.

Answer: Thank you for your suggestion. We added the Bader charge of every Co single atom in the revised Supporting Information (Supplementary Table 6).

Supplementary Table 6. Bader charge of various x -Co₁-N₃O₁ models.

Subs	1-Co ₁ -N ₃ O ₁	2-Co ₁ -N ₃ O ₁	4-Co ₁ -N ₃ O ₁
------	--	--	--

Bader charge (e)	0.8278	0.8175	0.7974
		0.8146	0.7952
			0.7969
			0.7933
Average (e)	0.8278	0.8161	0.7957

9. What's the difference about Co NPs for alkene epoxidation?

Answer: Thank you for your suggestion. We used DFT calculations to understand the trans-stilbene epoxidation reaction pathways of Co NPs. Both XRD and HRTEM characterizations revealed that Co NPs exposed with Co(111) crystal faces (Supplementary Fig. 38-39). Therefore, we used Co(111) to represent Co NPs/NC sample for calculations. As shown in Supplementary Fig. 40, an O₂ molecule was adsorbed and dissociated to O* on Co(111) with a high exothermic of -4.56 eV (III). The SB molecule was then activated and converted to SBO with an energy barrier of 1.21 eV (IV-V). By overcoming another energy barrier of 1.36 eV (V to VI), the remaining oxygen can be transformed to a new one-coordinated oxygen atom and subsequently participate in the next step of the SB epoxidation cycle. The highest energy barrier for SBO production reached to 1.78 eV (VII to VIII). It can be seen that the energy barrier on Co NPs was much higher than on Co1/NOC-x samples, resulting in the lowest reaction rate in trans-stilbene epoxidation, which was consistent with experimental results.

We added these discussions to the revised manuscript.

Supplementary Fig. 40 Energy profiles of trans-stilbene epoxidation reaction on Co(111) of Co NPs/NC models. Inset: the configurations of intermediates.

10. Fig. 1f is not visible, please verify it.

Answer: Thank you for your reminding. We updated the Fig. 1f in the revised manuscript.

REVIEWER COMMENTS

Reviewer #1 (Remarks to the Author):

The authors have made substantial progress to address reviewers' comments. The manuscript may be publishable after my remaining concerns are addressed:

1. Can the authors circle the location of CoNO sites in the HAADF-STEM images of 5.4 wt%, 10.9 wt%, and 21.2 wt% Co SAC samples (Fig 2a, 2b and 2c)?

2. The relative energies of II and III in Fig 5c are confusing for me. According to the mark of x-axis, II refers to $\text{Co1} + *O_2$, and III refers to $\text{Co1} + *O_2 + *SB$ (* denotes adsorbed state). Why the total adsorption energy (by the way, what is the y-axis, free energy, or electronic energy?) of O_2 in 4-Co1 SAC is so close to those in 1-Co1 and 2-Co1 SACs, given that there are four chemically adsorbed O_2 in 4-Co1, while 1 and 2 in 1-Co1 and 2-Co1. For *SB, why the adsorption energy in 4-Co1 is substantially smaller than that in 1-Co1 and 2-Co1?

3. I appreciate the authors' effort to calculate other pathways producing epoxy benzene and phenol, as collected in Fig S37, which aims to address my previous comment 7, "It is also surprising for me that the trans-stilbene (SB) adsorbed on the catalyst would not interact with two adjacent Co sites simultaneously, considering SB is large enough to cover two Co sites (Fig 5c)". In fact, I didn't suggest that O_2 would attach other C-C bond on the benzene rings than the C=C double bond connecting two benzene ring. What I meant was the SB molecule may binds to two Co sites simultaneously so that the binding energy can be very different from binding to only one Co site. After revisiting Fig 5, I found SB indeed binds to more than one $*O_2$. The isometric projection is not clear enough to show the binding configuration. They may provide an additional top view to make it easier to understand.

4. The answer to my previous comment 3 seems to be a little off the point. The authors did not build a connection between the change of electronic structure of Co due to the increase of active site density (Figure 3b, 3c and 3d) with the enhancement of catalytic activity ((Figure 4c) in their answer. Why does more Co atoms lead to lower spin moment of each Co and higher d-band center that are favorable for O_2 activation? How are TOF numbers obtained, by experiment or theoretical calculation?

5. In Line 282, the authors said "The larger variations in spin moment of O_2 in 4-Co1-N3O1 before and after oxygen adsorption" without showing the data of spin moment of O_2 before and after adsorption.

Reviewer #2 (Remarks to the Author):

After reviewing the authors' responses to my questions, I recommend publishing this manuscript in its

current form.

Reviewer #3 (Remarks to the Author):

The authors have addressed my concerns.

Response to reviewers

Reviewer #1

The authors have made substantial progress to address reviewers' comments. The manuscript may be publishable after my remaining concerns are addressed.

Answer: Thank you very much for your comments and valuable suggestions, which are helpful for us to improve the quality of manuscript. We revised the manuscript carefully according to your advice.

Question 1. Can the authors circle the location of CoNO sites in the HAADF-STEM images of 5.4 wt%, 10.9 wt%, and 21.2 wt% Co SAC samples (Fig 2a, 2b and 2c)?

Answer: Thank you for your suggestion. We have circled the location of Co single sites in the revised manuscript.

Fig. 2 (a-c) AC HAADF-STEM images of 5.4 wt%, 10.9 wt%, and 21.2 wt% Co SAC samples. Co single atoms are marked in red circles.

Question 2. The relative energies of II and III in Fig 5c are confusing for me. According to the mark of x-axis, II refers to $\text{Co1} + *O_2$, and III refers to $\text{Co1} + *O_2 + *SB$ (* denotes adsorbed state). Why the total adsorption energy (by the way, what is the y-axis, free energy, or electronic energy?) of O_2 in 4-Co1 SAC is so close to those in 1-Co1 and 2-Co1 SACs, given that there are four chemically adsorbed O_2 in 4-Co1, while 1 and 2 in 1-Co1 and 2-Co1. For *SB, why the adsorption energy in 4-Co1 is substantially smaller than that in 1-Co1 and 2-Co1?

Answer: Thank you for your comments. First, the adsorption energy in revised Fig. 5d is the free energy, and we have corrected it.

Second, in order to make the reaction pathways for three substrates be consistent, we only focus on the O_2 adsorption on the reactive site in Fig. 5d. It means that in the configurations I of 2-Co1-N₃O₁ and 4-Co1-N₃O₁, the adjacent Co sites were already covered by O_2 , and these O_2 maintained in an adsorbed state through all the reaction pathways. So the adsorption energy of O_2 (step I to step

II, free energy) in 4-Co₁-N₃O₁ is just the adsorption energy of one O₂ on the reactive site, rather than the total adsorption energy of four O₂ on the entire substrate. We also added an illustration of configurations step I in the revised Fig. 5d to avoid the possible misunderstanding.

Fig. 5 (d) Energy profiles of trans-stilbene epoxidation reaction over 1-Co₁-N₃O₁, 2-Co₁-N₃O₁ and 4-Co₁-N₃O₁ models. Inset: the configurations of intermediates of 4-Co₁-N₃O₁.

For *SB, the electronic charge density of adsorbed O₂ was found to play a significant role in the SB's subsequent adsorption on the catalyst substrate. As shown in Fig. 5c, more electrons were transferred from Co to O₂ in 4-Co₁-N₃O₁ (0.44 e vs. 0.38 e in 2-Co₁-N₃O₁ and 0.36 e in 1-Co₁-N₃O₁). Consequently, the larger electronic density in O₂ weakened its affinity to the C=C bond of SB with abundant π electrons, which led to the smaller adsorption energy of SB in 4-Co₁-N₃O₁. The attenuated interaction between O₂ and SB was also reflected in the difference of the C-O distances between them (2.48 Å in 4-Co₁-N₃O₁, 2.45 Å in 2-Co₁-N₃O₁, and 2.44 Å in 1-Co₁-N₃O₁). These may be the reasons why the adsorption energy of SB on 4-Co₁-N₃O₁ is weaker than on 1-Co₁-N₃O₁ and 2-Co₁-N₃O₁.

Question 3. I appreciate the authors' effort to calculate other pathways producing epoxy benzene and phenol, as collected in Fig S37, which aims to address my previous comment 7, "It is also surprising for me that the trans-stilbene (SB) adsorbed on the catalyst would not interact with two adjacent Co sites simultaneously, considering SB is large enough to cover two Co sites (Fig 5c)". In fact, I didn't suggest that O₂ would attach other C-C bond on the benzene rings than the C=C double bond connecting two benzene ring. What I meant was the SB molecule may binds to two Co sites simultaneously so that the binding energy can be very different from binding to only one Co site. After revisiting Fig 5, I found SB indeed binds to more than one *O₂. The isometric projection is not clear enough to show the binding configuration. They may provide an additional top view to make it easier to understand.

Answer: Thank you very much for your advice. According to your suggestion, we checked all the situations that SB interacts with two (and more than two) adjacent Co sites simultaneously. It was found that, if the center of SB (the double bond between two benzene rings) was bound to a Co site, this SB molecule would not closely contact with other adjacent Co sites simultaneously. As shown in Fig. R1, the distance of nearest three Co sites is 14.8 Å, while SB has a size of only 11.6 Å. So the minimum distance between terminal H and O in adjacent Co sites is more than 3.3 Å, which is significantly larger than the common O-H distance in an adsorbed state. As a result, SB could only interact with two adjacent Co sites simultaneously by attaching two Co sites with its two benzene rings (Supplementary Fig. 37) with much larger energy barriers.

We are sorry that we didn't clarify it clearly in last response. Besides, we have optimized the adsorption configuration of SB at one Co site, as shown in Fig. 5c ($\text{Co}_1 + \text{O}_2^* + \text{SB}^*$, step III). We also provided a top view of the intermediate configurations of $4\text{-Co}_1\text{-N}_3\text{O}_1$ in Supplementary Fig. 35, which showed clearly that SB binds to one $^*\text{O}_2$.

Fig. R1. The schematic sizes of SB molecule and $4\text{-Co}_1\text{-N}_3\text{O}_1$ support.

Supplementary Fig. 35 Top view of the intermediate configurations of $4\text{-Co}_1\text{-N}_3\text{O}_1$.

Question 4. The answer to my previous comment 3 seems to be a little off the point. The authors did not build a connection between the change of electronic structure of Co

due to the increase of active site density (Figure 3b, 3c and 3d) with the enhancement of catalytic activity (Figure 4c) in their answer. Why does more Co atoms lead to lower spin moment of each Co and higher d-band center that are favorable for O₂ activation? How are TOF numbers obtained, by experiment or theoretical calculation?

Answer: Thank you for your comment. First, we built a connection between the electronic structure of Co atom and catalytic activity. As shown in Fig. 5b, we can find that it presents a positive correlation between the catalytic activity and average Bader charge of Co atoms, where a sharply enhancement of TOF value realizes with the decrease of Bader charge, suggesting that the Bader charge might be a key descriptor in trans-stilbene epoxidation.

Second, it can be seen that the splitting of the d-orbitals shows distinct difference close to the Fermi energy level (Supplementary Fig. 23). The Co 3d orbitals of 1-Co₁-N₃O₁ is significantly split and more states are pushed above the Fermi energy level compared to that of 4-Co₁-N₃O₁, resulting in higher spin moment and lower d-band center. Meanwhile, this implies that more charge transfer should happen from the Co to the support, consistent with XANES, XPS and Bader charge analysis. This tendency is consistent with results in literature (e.g. *Nat. Catal.* 2022, 5, 485–493). Moreover, Schnöckel et al. reported the decreased spin would result in higher activity for O₂ activation (*Science*, 2008, 319, 438-442). Therefore, in our manuscript, 4-Co₁-N₃O₁ with lowest spin moment displayed the highest activity.

Third, the turnover frequencies (TOFs) of these Co₁/NOC-x samples were calculated according to the experiment results, as following equation (1),

$$\text{TOF (h}^{-1}\text{)} = \left(1 - \frac{n_t}{n_0}\right) \times \frac{n_0}{n_{\text{Co}} \times t} \quad (1)$$

Where n_0 is the initial mole of trans-stilbene, n_t is the mole of trans-stilbene at time, n_{Co} is the total mole of Co in the sample, and t is the reaction time in hour.

We added above discussions in the revised manuscript.

Fig. 5 (b) The relationship between Bader charge and catalytic activity.

Question 5. In Line 282, the authors said "The larger variations in spin moment of O₂ in 4-Co₁-N₃O₁ before and after oxygen adsorption" without showing the data of spin moment of O₂ before and after adsorption.

Answer: Thank you for your suggestion. We added the spin moments in total O₂ molecule before and after adsorption in the table below, as well as the variations (ΔM).

Supplementary Table 10. The spin moments of O atoms in different x-Co₁-N₃O₁ models before and after adsorption.

Subs	1-Co ₁ -N ₃ O ₁	2-Co ₁ -N ₃ O ₁	4-Co ₁ -N ₃ O ₁
Before adsorption (μ_B)	1.616		
After adsorption (μ_B)	0.838	0.812	0.770
ΔM (μ_B)	0.778	0.804	0.846

Reviewer #2

After reviewing the authors' responses to my questions, I recommend publishing this manuscript in its current form.

Answer: We really appreciate you for your positive opinion on our work as well as previous helpful suggestions.

Reviewer #3

The authors have addressed my concerns.

Answer: We really appreciate you for your positive opinion on our work as well as previous helpful suggestions.

REVIEWERS' COMMENTS

Reviewer #1 (Remarks to the Author):

My comments have been addressed. Now it is publishable.

Response to reviewers

Reviewer #1

My comments have been addressed. Now it is publishable.

Answer: We really appreciate you for your support on our work as well as previous helpful suggestions.